# Extreme atmospheric rivers in a warming climate

**Shuyu Wang[1], Xiaohui Ma [1,2] ✉, Shenghui Zhou[2], Lixin Wu [1,2], Hong Wang [1,2], Zhili Tang[1], Guangzhi Xu[3], Zhao Jing [1,2], Zhaohui Chen [1,2] & Bolan Gan [1,2]**

Extreme atmospheric rivers (EARs) are responsible for most of the severe precipitation and disastrous flooding along the coastal regions in midlatitudes. However, the current non-eddy-resolving climate models severely underestimate (~50%) EARs, casting significant uncertainties on their future projections. Here, using an unprecedented set of eddy-resolving high-resolution simulations from the Community Earth System Model simulations, we show that the models' ability of simulating EARs is significantly improved (despite a slight overestimate of ~10%) and the EARs are projected to increase almost linearly with temperature warming. Under the Representative Concentration Pathway 8.5 warming scenario, there will be a global doubling or more of the occurrence, integrated water vapor transport and precipitation associated with EARs, and a more concentrated tripling for the landfalling EARs, by the end of the 21st century. We further demonstrate that the coupling relationship between EARs and storms will be reduced in a warming climate, potentially influencing the predictability of future EARs.

Atmospheric rivers (ARs) are synoptic high water vapor transport corridors usually occurring ahead of the cold front of an extra-tropical cyclone in midlatitudes[1,2]. Being important conveyors between oceanic evaporation and continental precipitation, they are responsible for most of the precipitation extremes and flooding events when making landfall along the coastlines of like North America and Europe[3,4]. Over 85% of flood events along the western coast of the United States (US) are related to ARs while the absence of ARs may increase the occurrence of droughts up to 90%[5,6], exerting a substantial socio-economic impact in the affected areas. Aside from the impacts on water extremes, ARs are also closely related to storms and wind extremes[7,8] and are one of the most important sources of climate hazards in the midlatitude[9,10]. Although the impacts of ARs reach globally, the majority of current studies focus on the most AR-impacted regions like the west coast of North America and Europe[11–17]. Understanding ARs' response to anthropogenic warming from a global perspective is highly demanding and crucial for the prediction of weather systems and hydrological extremes and the preparation of potential threats associated with them[15,18,19].

## Results

### Observed and simulated extreme ARs

The focus of the study is extreme ARs (EARs) classified by their associated integrated water vapor transport (IVT) intensities (with the maximum AR IVT exceeding 1250 kg/m/s, see Methods for detailed definition and detections) and are claimed to be primarily hazardous[20]. It has been well established that ARs will occur more frequently with higher intensity and heavier precipitation in a warming climate because of the greater availability of water vapor in the atmospheric column according to the Clausius–Clapeyron relationship[11,15,18,21–23]. However, existing studies about ARs' response to global warming are largely based on non-eddy-resolving low-resolution climate simulations[22,24] and these climate models show systematically negative frequency/intensity biases in simulating EARs (Fig. 1), leading to uncertainties in their future AR projections. Specifically, compared to

[1]Frontiers Science Center for Deep Ocean Multispheres and Earth System and Key Laboratory of Physical Oceanography, Ocean University of China, Qingdao, China. [2]Laoshan Laboratory, Qingdao, China. [3]College of Global Change and Earth System Science, Beijing Normal University, Beijing, China. ✉ e-mail: maxiaohui@ouc.edu.cn

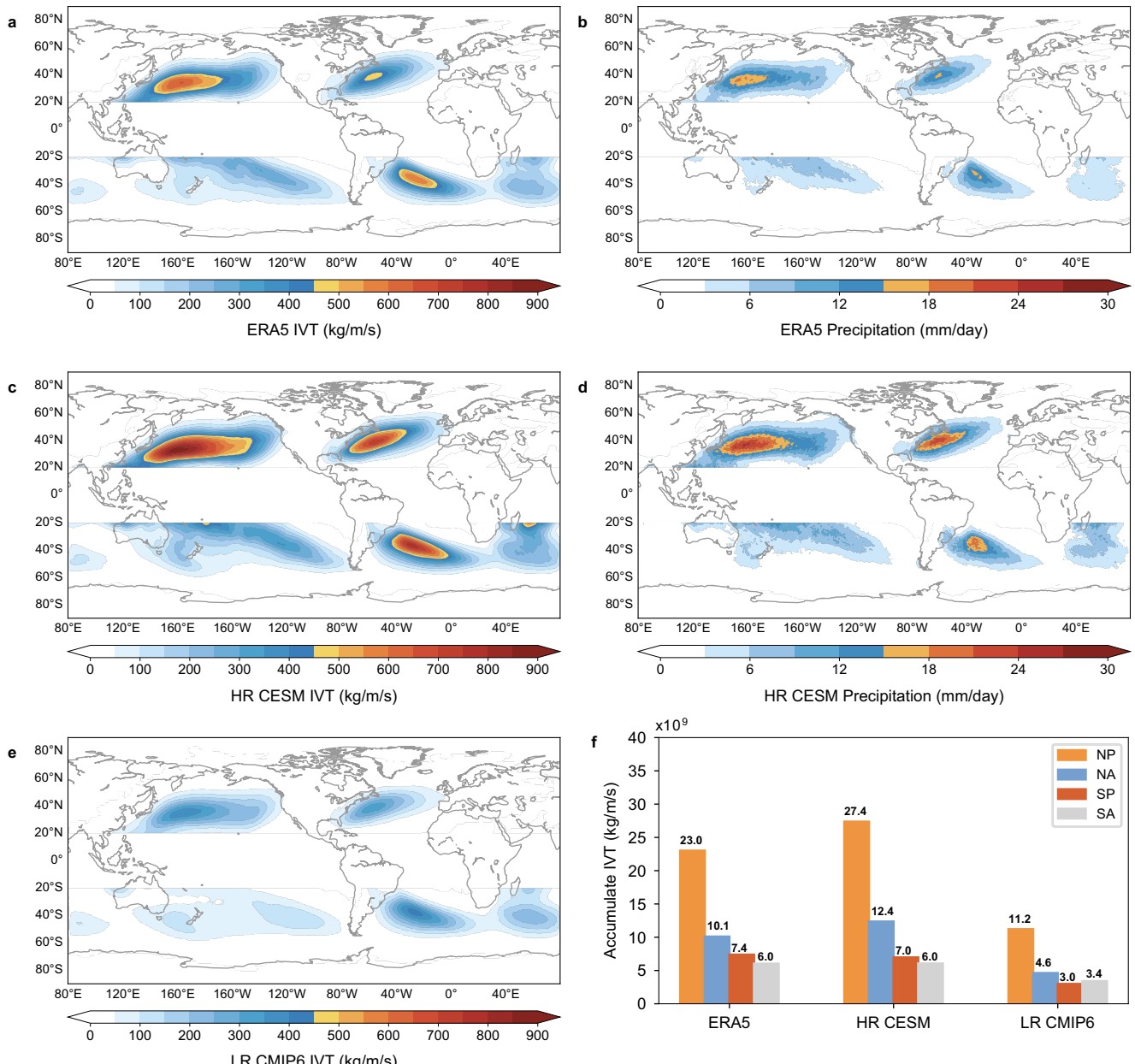

**Fig. 1 | Observed and simulated extreme atmospheric rivers (EARs).** Normalized accumulated EAR integrated water vapor transport (IVT, kg/m/s) in boreal winter season (ONDJFM) in ERA5 reanalysis (**a** the fifth generation European Center for Medium-Range Weather Forecasts atmospheric reanalysis[25]), HR-CESM (**c** High Resolution Community Earth System Model) and LR-CMIP6 (**e** Low Resolution Coupled Model Intercomparison Project Phase 6) during 1979–2005. **b**, **d**, as for **a**, **c**, but for precipitation (mm/day). EAR IVT (kg/m/s) averaged in AR-active regions (red boxes outlined in Fig. 2d) for ERA5 reanlysis, HR-CESM and LR-CMIP6 during 1979–2005 (**f**). Calculation details of EAR IVT and precipitation are given in Methods. Source data are provided as a Source Data file.

the ERA5 reanalysis (the fifth generation European Center for Medium-Range Weather Forecasts atmospheric reanalysis[25]), the normalized accumulated IVT associated with EARs in boreal winter season is underestimated by over 50% in AR-active regions in low-resolution (LR) CMIP6 simulations (the Coupled Model Intercomparison Project Phase 6, Methods) (Fig. 1a, e, f). The boreal winter season is chosen since the Northern Hemisphere (NH) EARs are strongest in winter and the seasonality of the Southern Hemisphere (SH) EARs is relatively weak. The normalized accumulated IVT takes account of the combined impact of the frequency and intensity of EARs (see Methods for detailed calculation). Note that EAR IVTs along the coasts are less evident than that in the western boundary current regions, due to lower occurrence frequency and weaker intensity of EARs (Fig. S1), which may be related to the lower background IVT over land than the

warm ocean. Detailed analyses of high-risk landfalling EARs along the coasts are shown in a later section.

The bias problem exists also in HighResMIP (High Resolution Model Intercomparison Project) simulations and the underestimates of EARs remain as high as ~40% (Fig. S2). This is likely due to the restricted eddy-resolving capability of the ocean component of High-ResMIP. Although the resolution of atmospheric models in currently available HighResMIP that provides high frequency outputs to detect ARs largely falls within the scope of eddy-resolving, the finest ocean resolution is ~0.25° and fails to fully capture mesoscale oceanic eddies (10 s ~ 100 s km) (Methods).

Very recently, an unprecedented set of multi-century high-resolution (HR) (~0.25° for the atmosphere and ~0.1° for the ocean) Community Earth System Model (CESM) simulations which are

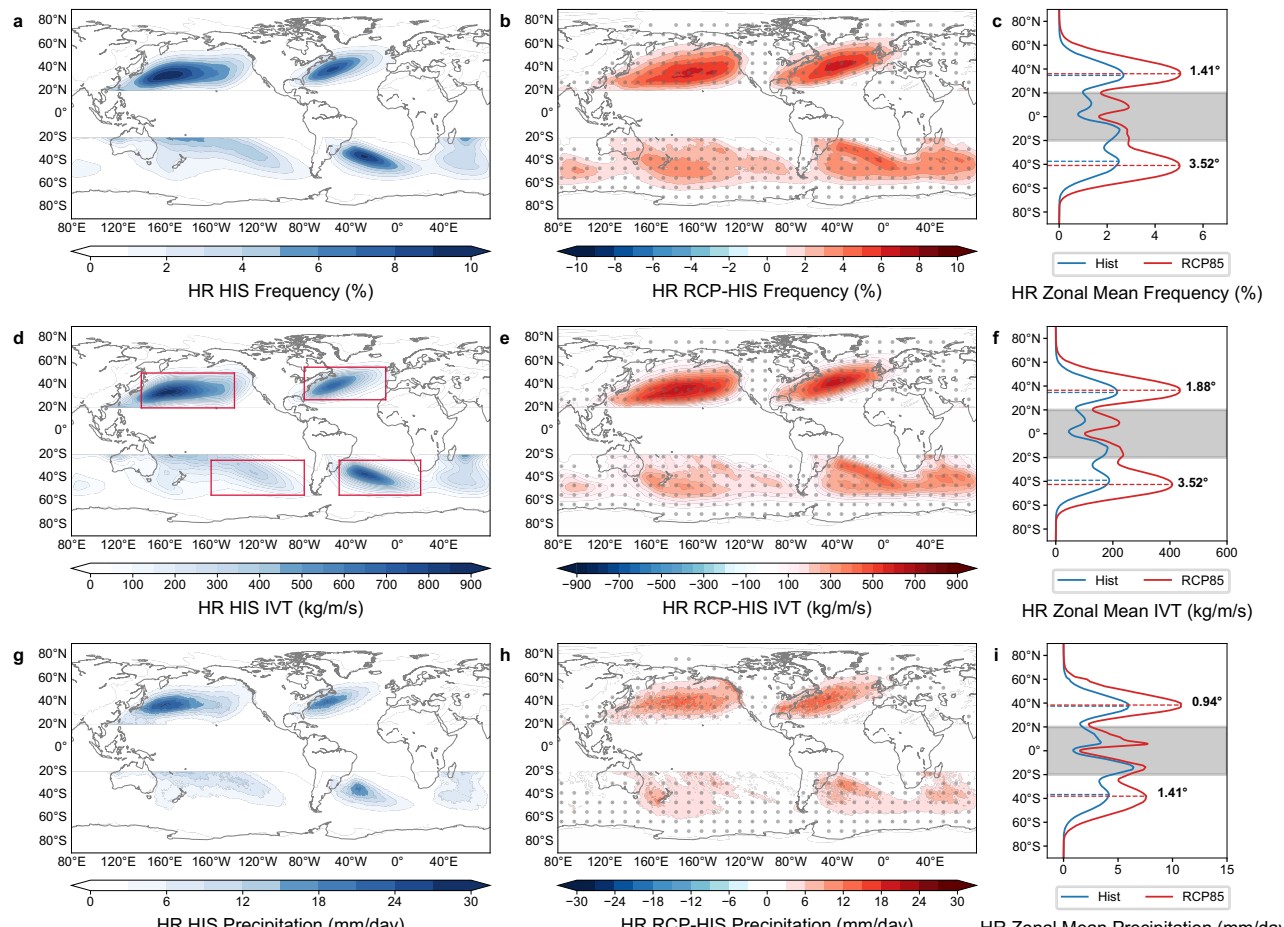

**Fig. 2 | Extreme atmospheric river (EAR) response to anthropogenic warming in High Resolution Community Earth System Model (HR CESM).** EAR occurrence frequency (%, **a**), normalized accumulated integrated water vapor transport (**d**, IVT, kg/m/s) and precipitation (**g**, mm/day) simulated in historical simulations (HR-HIS, 1956–2005) and the difference of that between future simulations (HR-RCP, 2051–2100) and HR-HIS (**b**, **e**, **h**). The difference above 95% confidence level based on a two-sided Student's test is shaded by gray dots. The zonally averaged EAR occurrence frequency (%, **c**), IVT (**f** kg/m/s) and precipitation (**i** mm/day) in HR-HIS (blue) and HR-RCP (red). The dotted lines and the numbers in **c**, **f**, **i** indicate the position of maximum EARs and the latitudinal shifts between HR-RCP and HR-HIS. The tropics ([20°S-20°N]) is blocked out to focus on EARs in the extratropics. Red boxes in Fig. 2d outline four EAR-active region in the North Pacific (NP), North Atlantic (NA), South Pacific (SP), South Atlantic (SA) to compute the global averaged value. Source data are provided as a Source Data file.

"eddy-resolving" for both the atmosphere and the ocean were available[26] (See Methods for model details). We find that both the accumulated IVT and precipitation associated with EARs are reproduced reasonably well in the HR CESM (Fig. 1a–d, f). Although the amplitude of EARs is slightly overestimated (~10%) (Fig. 1f), the simulated EARs are significantly improved in HR CESM compared to LR CMIP6 and HighResMIP. The improvement of EARs is independent of the resolution of IVT data as EARs of similar amplitude are observed when regridding the HR IVT onto the LR grid (figure not shown). Further comparison of simulated EARs between HR and a set of parallel LR CESM simulations (Methods) indicates that the better-resolved EARs in HR CESM are resulted from both thermodynamic and dynamic improvements with a higher contribution from the latter (Fig. S3). The increase of spatial resolution not only enhances water vapor[26] but also modifies the dynamic field related to extratropical cyclones or atmospheric circulations that are favorable for EAR formation, although detailed processes that contribute to the improvement of the EAR are multifaceted. Overall, the above results suggest that eddy-resolving climate models for both the atmosphere and the ocean are required to get more realistic simulations of EARs. With decent simulated EARs, HR CESM provides a unique opportunity to examine the response of EARs to anthropogenic warming that may be severely biased in previous climate simulations.

## Global response of extreme ARs to anthropogenic warming

Two studying periods with 6-hourly output in HR CESM, i.e., 1956–2005 from the historical simulation (HR-HIS) and 2051–2100 from the future simulation (HR-RCP) are chosen to examine the EARs response under the RCP8.5 warming scenario (Methods). Figure 2 shows the occurrence frequency, normalized accumulated IVT, and precipitation associated with EARs in HR-HIS and the corresponding differences between HR-RCP and HR-HIS. It is evident that more EARs are projected to occur across the globe and the occurrence frequency is almost doubled by the end of the 21st century (Fig. 2a–c). Correspondingly, the accumulated IVT and precipitation associated with EARs in HR-RCP are also more than two times those in HR-HIS (Fig. 2d–i). Furthermore, the position of maximum EAR occurrence in the future is projected to shift poleward with a more evident shifting (~3.5°) in the SH than in the NH (Fig. 2c, f) and the positional variation is related with the latitudinal change of storm tracks as shown later. Existing studies have noted that considerable uncertainties may be induced in AR statistics by different AR detection tools (ARDTs)[27]. Two additional ARDTs (Methods) are applied to verify the robustness of EAR projections in HR CESM. Although the amplitude of detected EARs varies among different ARDTs (Fig. S4), the projected EAR changes under global warming are highly consistent, all demonstrating a global doubling of EARs with similar spatial distribution in future climate.

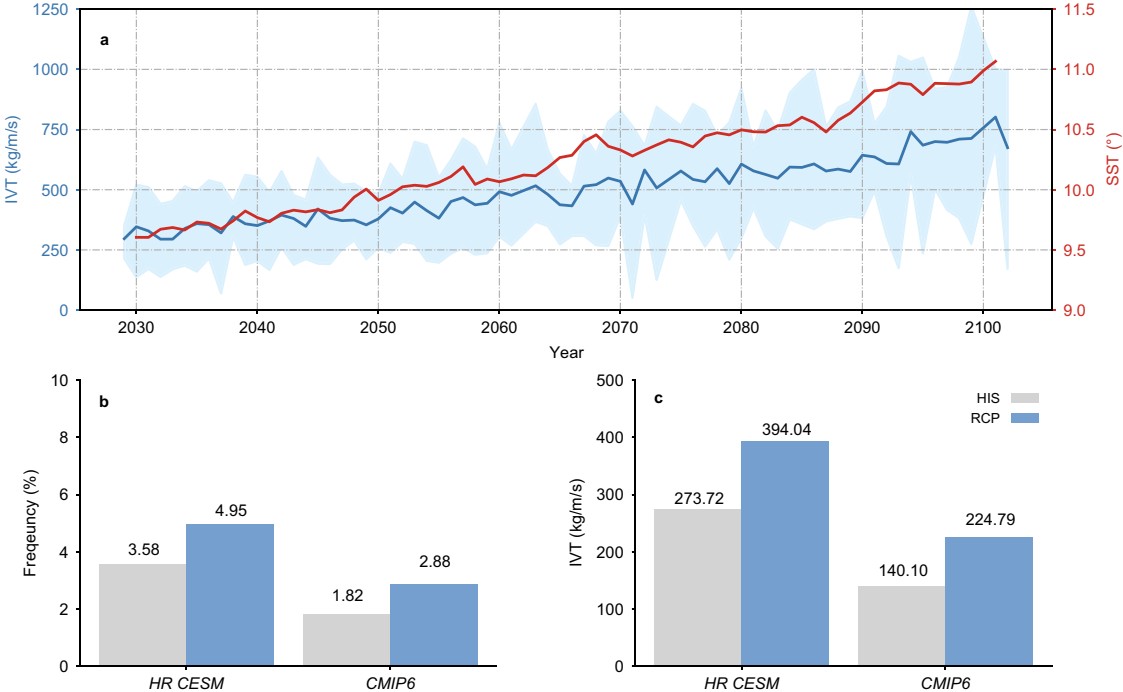

**Fig. 3 | Extreme atmospheric river (EAR) response to different warming levels.** Time series of global averaged accumulated EAR integrated water vapor transport (IVT, blue line) and sea surface temperature (SST, red line) from 2030 to 2100 in High Resolution Community Earth System Model (HR CESM) (**a**). The blue shading outlines the maximum and minimum accumulated EAR IVTs averaged in the North Pacific (NP), North Atlantic (NA), South Pacific (SP), and South Atlantic (SA). Global averaged EAR occurrence frequency (**b** %) and normalized accumulated IVT (**c** kg/m/s) in historical (1980–2000) and future (2030–2050) periods in HR CESM and CMIP6 (including LR CMIP6-Low Resolution Coupled Model Intercomparison Project Phase 6 and HighResMIP-High Resolution Model Intercomparison Project). The global averaged EAR IVT/frequency is computed by averaging the corresponding values in the red boxes outlined in Fig. 2d. Source data are provided as a Source Data file.

The sensitivity of EAR response to different warming levels is also evaluated. The result shows that the EARs increase almost linearly with temperature warming. The global averaged EAR-induced IVT is raised from 250 kg/m/s to 750 kg/m/s during 2030–2100, corresponding to a 1.5 °C sea surface temperature (SST) rising (Fig. 3a). The estimated increasing rate of EAR-related IVT during this period is 70 kg/m/s/ 0.2 °C per decade (~25% per decade in reference to the historical value). Additionally, in a lower-level warming period (2030–2050), EAR IVT reaches ~400 kg/m/s by the mid-century, which is 1.45 times the historical value but half the value of 2100 (Fig. 3c). The enhanced IVT is primarily attributed to the increased occurrence frequency of EARs (Fig. 3b). The projected EAR changes in CMIP6 (including HighResMIP) in the same period is examined and compared with HR CESM. Although the rising tendency of EARs with warming is captured, the absolute value of projected EARs is ~50% lower than those in HR CESM (Fig. 3b, c). The results indicate the severely biased EAR simulation in CMIP6 likely leads to further underestimates of future EARs and associated hydrological extremes under anthropogenic warming. However, it is worthwhile to note that the magnitude of the relative EAR change projected by LR CMIP6 is comparable to that in HR CESM. This implies that LR climate models may still provide valuable information about future EAR projections if the systematic model bias in their historical simulations is precisely known and corrected, as also noted in previous mean precipitation projections[28].

Diagnostic analyses are conducted to separate the thermodynamic and dynamic contributions to EAR changes following a previous study[29] (see Methods for details). The results indicate that the increase of EARs under warming is largely determined by thermodynamic change and agrees well with the overall water vapor rising in HR-RCP globally (Fig. S5). In contrast, the dynamic change is much weaker (~25% of the thermodynamic value) and acts to suppress AR increase especially in the NH, in line with the storm track changes

(Fig. S5). Moreover, the positional variations of EARs in both hemispheres shown in Fig. 2 are also consistent with the poleward shift of storm tracks (Fig. S5b). The predominant role of thermodynamic response in determining AR projection under warming has been widely discussed in many previous studies at regional scales[11,19,30] and the possible negative role of dynamic response in affecting AR changes has also been noted in the North Pacific and Mediterranean[29,31]. It is further proved in HR CESM that similar mechanisms can be applied at a global scale.

## Global response of landfalling extreme ARs to anthropogenic warming

ARs can induce extreme precipitation when making landfall, especially in high topography areas. Figure 4 shows the landfalling EAR response to global warming in the North Pacific (NP), North Atlantic (NA), and South Pacific (SP) in HR CESM (the number of EARs landfalling along the east coast of the South Atlantic is small and is not shown here). Consistent with the overall EAR changes, future landfalling EARs are also significantly enhanced with an even stronger amplification (Fig. 4a–f). Compared to HR-HIS, the accumulated IVT and precipitation associated with landfalling EARs in HR-RCP are generally increased by a factor of two in 2050–2100 (Fig. 4a–i). Also, it is evident that the landfalling EARs induced precipitation relies heavily on topographical lifting, and the precipitation enhancement in these areas can reach as high as 200–300% (Fig. 4g–i). The results suggest a higher amplification rate of landfalling EARs (~threefolds) than global EARs (~twofolds) in a future warming climate and it is these landfalling EARs that will cause severe hydrological hazards and directly impact the coastal regions.

Possible factors that contribute to the disproportionate increase of landfalling EARs under warming are further investigated. Examination of the evolution of landfalling EARs reveals a two-fold

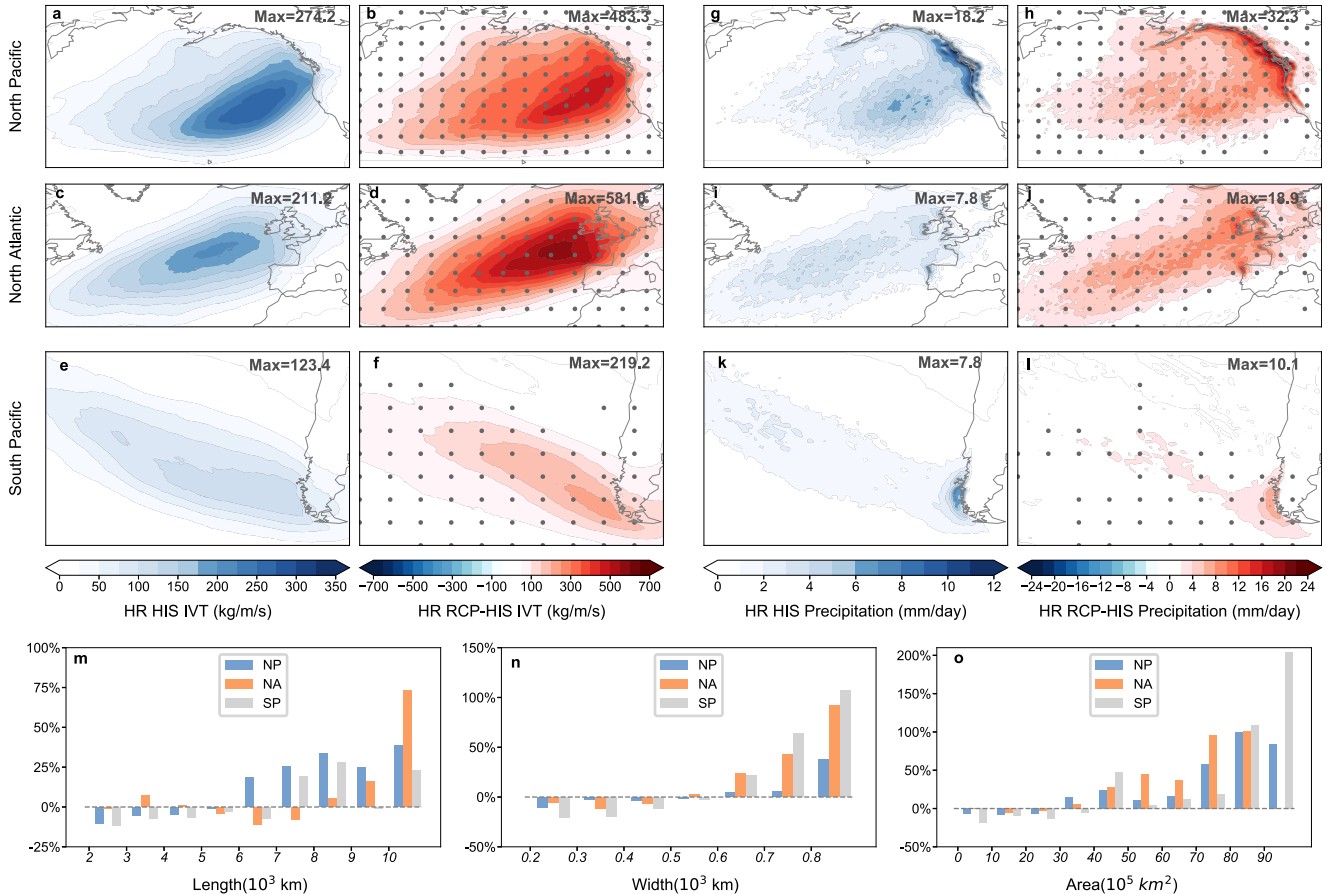

**Fig. 4 | Landfalling extreme atmospheric river (EAR) response to anthropogenic warming in High Resolution Community Earth System Model (HR CESM).** Normalized accumulated integrated water vapor transport (IVT, kg/m/s) associated with landfalling EARs in the North Pacific (NP) (**a**), North Atlantic (NA) (**c**) and South Pacific (SP) (**e**) simulated in historical simulations (HR-HIS) and the corresponding differences between future simulations (HR-RCP) and HR-HIS (**b**, **d**, **f**). **g–l** as for **a–f**, but for precipitation (mm/day). The maximum value for each figure is labeled on the top right. Calculation details of EAR IVT and precipitation are given in Methods. The difference above 95% confidence level based on a two-sided Student's test is shaded by gray dots. Probability distribution functions (PDFs) of relative differences of landfalling EARs' length (**m** $10^3$ km), width (**n** $10^3$ km) and area (**o** $10^5$ km$^2$) between HR-RCP and HR-HIS in reference to HR-HIS for the NP, NA and SP, respectively. Source data are provided as a Source Data file.

amplification in the genesis occurrence comparable with global EARs' change while the occurrence frequency comes to three-fold the historical value at landfalling (Fig. S6), implying a modification of EARs during the propagation. PDFs of AR characteristics indicate an overall lengthening and widening of landfalling EARs (Fig. 4m–o). The fractional increase of AR's length and width between HR-RCP and HR-HIS is up to 50–100%, leading to a corresponding growth of AR's size as high as 100–200%. The elongation and broadening characteristics of EARs are preferential for downstream extension and promote their interaction with land. Combined with the topographical lifting, more landfalling EARs are produced.

### Relationship between EARs and storms
The occurrence of ARs is closely related with extratropical cyclones (ECs)[32,33]. Over 80% of EARs are paired with ECs (see Methods for definition of paired AR-EC) in HR-HIS and the pairing relationship drops slightly (2% ~ 5%) in HR-RCP (Tab. S1). Composites of sea level pressure (SLP) and IVT associated with EARs show a clear low pressure system northwest (southwest) of EARs in the NH (SH) (Fig. 5a, e), and the intense IVT of EARs aligns nicely along the largest pressure gradient (contours in Fig. 5c, g). In HR-RCP, the EARs are enhanced with stronger IVT (Fig. 5b, f) but the intensity of the accompanied storms is reduced with a weaker SLP gradient (shading in Fig. 5c, g) and higher central SLP (Fig. 5b, f). The reduced storm intensity is more evident in the NH than the SH, again consistent with the storm track change

(Fig. S5). Specifically, the averaged SLP gradient around the maximum IVT of the EAR center is significantly weakened by ~20% (10%) in the NH (SH) (Fig. 5c, g). Meanwhile, the percentage of EARs pairing with weaker storm intensity (higher SLP) is raised as high as 50% in HR-RCP (Fig. 5d). The above results imply that future EARs tend to be paired with less intense storm systems. This is understandable given that the enhanced water vapor supply in a warming climate is likely to lower the requirement of the wind field for EAR formation.

Another modification of the pairing relationship between EARs and ECs is that future EARs tend to locate further away from the EC center with a growing distance between AR and storm centers (Fig. 5h), although the distance change is not as evident as the storm intensity response. Collectively, the above results tend to suggest there will be a reduced coupling between EARs and the strength of storms under global warming, and the development of future EARs will be less dependent on storm systems due to greater support from moisture supply. The high internal variability induced by synoptic storms is the primary source that undermines the predictability of EARs[34]. The reduced coupling between EARs and storms potentially suggests future EARs may become more predictable, although in-depth analyses are required to evaluate the prediction skill in the future.

### Discussion
The current non-eddy-resolving climate models (including LR CMIP6 and HighResMIP) severely underestimate EARs in their historical

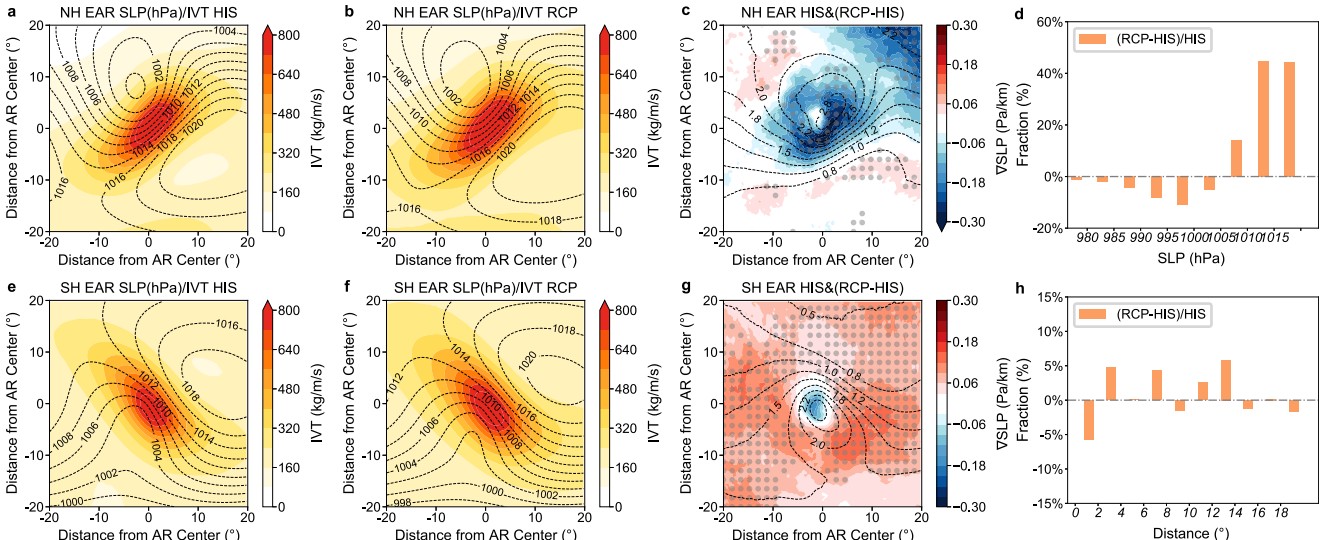

**Fig. 5 | Coupling relationship between extreme atmospheric rivers (EARs) and extratropical cyclones (ECs).** Composite of sea level pressure (SLP, hPa, contour) and integrated water vapor transport (IVT, kg/m/s, shading) associated with all EARs pairing with ECs in historical (**a** HR-HIS) and future (**b** HR-RCP) simulations in the Northern Hemisphere (NH). Composite of SLP gradient (∇SLP, Pa/km) associated with all EARs in HR-HIS (contour) and the corresponding difference between HR-RCP and HR-HIS (shading) (**c**) in the NH. **e**–**g** as for **a**–**c**, but for the Southern Hemisphere (SH). The difference above 95% confidence level based on a two-sided Student's test is shaded by gray dots. Probability distribution functions (PDFs) of relative difference of centered EC SLP (hPa) associated with EARs between HR-RCP and HR-HIS in reference to HR-HIS (**d**). **h** as for **d** but for the distance (°) between AR and EC center. Source data are provided as a Source Data file.

simulations and possibly future EAR changes under anthropogenic warming, which could lead to serious distortion of future climate adaption. With significantly improved EAR simulations, the eddy-resolving HR CESM offers a unique tool for providing more reliable EAR projections. EARs are projected to be doubled globally by the end of the 21st century under RCP8.5 warming scenario and a more concentrated tripling for the landfalling EARs is projected, compared to the last century. The higher amplification of landfalling EARs is attributed to the lengthening and broadening of EARs, which favors extended intrusion into the land. In a warming climate, the thermodynamic control on EAR genesis is so strong that EARs become less relevant with storms. The reduced coupling between EARs and storms suggests a potential increase of future EARs' predictability.

There are still uncertainties that cannot be excluded in EAR projections in the study. It is possible the increase of EARs projected by HR CESM may be overestimated to some extent, and more eddy-resolving climate models are needed to evaluate the uncertainties in the future. Also, ERA5 IVT itself may be biased compared to drop-sonde observations as reported in a recent study[35], which requires more drop-sonde observations at a global scale to reduce model validation uncertainties. Moreover, it needs to point out that the primarily hazardous EARs (IVT > 1250 kg/m/s) are chosen in this study according to a recent classification[20] and the amplitude of EAR projections may vary with different EAR definitions. Sensitivity tests of EARs defined by different IVT thresholds (the 75th and 90th percentile values, Fig. S7) suggest that a higher IVT threshold is likely to generate fewer EARs but more intense EARs' increase to global warming.

## Methods
### CESM simulations
The HR CESM models are developed by National Center for Atmosphere Research (NCAR) and employ ~0.25° atmosphere and land components (the spectral element dynamic core, SE-dycore; Community Atmosphere Model version 5, CAM5; the Community Land Model version 4, CLM4) and ~0.1° ocean and sea ice components (the Parallel Ocean Program version 2, POP2; the Community Ice Code version 4, CICE4)[26,36]. The simulations consist of a 500-year preindustrial control simulation and a 250-year historical and future

climate simulation from 1850–2100. Historical forcing is applied from 1850 to 2005 while RCP8.5 warming forcing (a high-level greenhouse gas concentration) is applied from 2006 to 2100[37–39]. To allow a complete model adjustment to the switched forcing and to maximize the warming effect, a later period (2051–2100) in future simulations and a corresponding historical period (1956–2005) were chosen. The CESM version used is CESM1.3 with updated microphysics, radiation, gravity wave scheme and tuned dust and soil erodibility[26], which produces improved jet stream and cloud simulations.

To investigate the attributable factors of improved EAR simulations with the increased model resolution, a set of parallel LR CESM simulations (~1° for both the atmosphere and the ocean) are conducted. The model settings are identical except for resolution and some retuned parameters to achieve top-of-atmosphere balance (See Chang et al. 2020[26] for details). Like LR CMIP6, EARs simulated in LR CESM are substantially lower than that in HR CESM (Fig. S3), confirming that non-eddy-resolving models show poor ability in capturing EARs. The respective contribution of thermodynamic and dynamic changes to the improved EAR simulation in HR CESM compared to LR CESM, is also shown in Fig. S3.

### CMIP6 and HighResMIP simulations
High frequency (6-hourly/daily) IVT or three-dimensional u, v, q are needed to identify ARs. Four HighResMIP simulations satisfying the requirement are founded, i.e., EC-Earth3P-HR (0.5° atmosphere and 0.25° ocean), HadGEM3-GC31-HM (0.5° atmosphere and 0.5° ocean), MPI-ESM1-2-XR (0.5° atmosphere and 0.5° ocean), CMCC-CM2-VHR4 (0.25° atmosphere and 0.25° ocean). One of these HighResMIP simulations has a comparable atmospheric resolution (0.25°) to HR CESM, but none of them are eddy-resolving in the ocean component (the finest ocean resolution is 0.25°).

Four paring LR (~1° atmosphere and ocean) CMIP6 simulations were selected, i.e., EC-Earth3P (1° atmosphere and 1° ocean), HadGEM3-GC31-MM (1° atmosphere and 1° ocean), MPI-ESM1-2-HR (1° atmosphere and 0.5° ocean), CMCC-CM2-HR4 (1° atmosphere and 0.25° ocean). All the selected models include historical and future simulations (under RCP8.5 warming scenario) from 1950 to 2050, and

EARs in the overlapping period with ERA5 and HR CESM are analyzed and compared (Figs. 1 and 3, and Fig. S2).

### AR detection, definition of EARs and calculation of EAR IVT

The primary ARDT used in this study is based on the classic definition of ARs[20], which searches for long narrow features of IVT anomalies exceeding 250 kg/m/s (hereafter IVT250) using 6-hourly IVT $(\frac{1}{g} \int_{1000hpa}^{Ptop} (\sqrt{(uq)^2 + (vq)^2})$. Additional geometric restrictions applied include a minimum length requirement of 800 km, a minimum length/width ratio of 2 and a minimum isoperimetric quotient ratio of 0.7 following a previous study[40]. A minimum 24 h duration restriction is also applied to the tracking to exclude short-lived moisture filaments. To consider the mean state change of background IVT under warming, IVT anomalies in historical and future periods are defined as 6-hourly deviations from the 50-year historical and RCP8.5 climatological means, respectively. Landfalling ARs are defined as if the outer edge of an AR intersects the coastlines of continents. In both hemispheres, we focus on extratropical ARs within 20° to 80° latitudinal bands.

To test the result sensitivity on ARDTs, another two ARDTs from ARTMIP–IVT85%[41,42] and IPART are applied in HR-CESM (Fig. S4). The IVT85% method[40] uses the 85th percentile flexible IVT threshold to define ARs and the IPART[40] is based on an image processing algorithm and is IVT threshold-free.

An EAR is defined as if the maximum AR IVT exceeds 1250 kg/m/s (Category 4 and 5 ARs as classified by Ralph et al. 2019[20]) and is claimed to be primarily hazardous. Considering the large regional disparity of EARs, the application of regional-specific IVT thresholds to define EARs may help to well represent AR features in specific regions. However, it will also induce unfair comparisons of EARs among different regions. To avoid this, a ubiquitous IVT threshold is applied throughout the globe in this study, which allows a fair comparison of global EARs among different basins.

Accumulated IVT is computed to get a full assessment of the combined impact of the frequency and intensity change of EARs. To improve the readability, the accumulated IVT is then normalized by a ubiquitous EAR occurrence number to represent the typical EAR strength. The ubiquitous EAR occurrence number is derived as the maximum occurrence of climatological EARs in HR-HIS, corresponding to (49) 72 (landfalling) EARs per winter. Specifically, the EAR IVT at each grid is computed as $\frac{\sum_{EAR=1}^{n} IVT}{N}$ (where IVT is the intensity of an individual EAR, n is the total number of EARs at the grid point, N is the maximum EAR occurrence number used in the normalization). Note that although the individual EAR IVT is all above 250 kg/m/s, EARs are distributed over space and do not perfectly align. Therefore, it is possible that only a fraction of EARs overlapped at any given computing grid point (n < N), leading to reduced IVT values (Figs. 2d and 3c) compared with individual EAR IVT. A similar calculation is applied for precipitation.

### EC detection and paired AR-EC

ECs are detected using SLPa[32,43,44] and SLPa are derived by applying a temporal (15-day high-pass) and a spatial (1°x1° low-pass) filtering to target the synoptic storm features[45]. The EC center is located at the SLPa minimum and the outer edge of an EC is defined as the utmost closed contour of SLPa without including additional SLPa minimum inside. An EC is paired with an AR if the EC center locates within a 25°x25° box around the AR center (the IVT maximum of an AR)[32,45].

### Thermodynamic and dynamic separation

The separation of thermodynamic and dynamic AR responses to global warming follows a previous study[29]. Assuming $V_1(x,y,t)Q_1(x,y,t)$ and $V_2(x,y,t)Q_2(x,y,t)$ represent IVT in HR-HIS and HR-RCP, respectively. At each grid point and each time step, $V_1(x,y,t)Q_1(x,y,t)$ is rescaled by a factor of $\frac{\bar{Q}_2}{\bar{Q}_1}(x,y)$, where $\bar{Q}_1$ and $\bar{Q}_2$ are the 50-year climatological mean of integrated water vapor (IWV) in HR-HIS and HR-RCP. By rescaling, the historical IWV is replaced by the amplified IWV in HR-RCP and is referred to as HR-RCP-Q while the wind is kept the same. Thus, comparison of EARs between HR-HIS and HR-RCP-Q can give an estimate of EARs changes due to water vapor (thermodynamic contribution). The dynamic contribution is then taken as the residual between total and thermodynamic EAR changes. It should be noted that this rescaling approach only provides an estimated contribution of the two effects by assuming the total water vapor change between HR-HIS and HR-RCP can be largely reproduced by their mean values and it is verified that the assumption is fairly accurate with less than 10% error introduced[29]. The separation of thermodynamic and dynamic in EAR IVT difference between HR CESM and LR CESM is conducted similarly by assuming $V_1(x,y,t)Q_1(x,y,t)$ and $V_2(x,y,t)Q_2(x,y,t)$ represent IVT in HR-HIS and LR-HIS, respectively.

## Data availability

Observed IVT used to detect ARs in ERA5 can be downloaded from https://www.ecmwf.int/en/forecasts/datasets/reanalysis-datasets/era5. The HR and LR CESM simulations can be achieved from http://ihesp.qnlm.ac and https://ihesp.github.io/archive/products/ds_archive/Sunway_Runs.html. The CMIP6 and HighResMIP simulations can be downloaded from https://esgf-data.dkrz.de/search/cmip6-dkrz/. Source data are provided with this paper and also can be obtained via https://doi.org/10.6084/m9.figshare.22631143.

## Code availability

The AR detection code is available at https://doi.org/10.21105/joss.02407 and ARTMIP (https://www.cgd.ucar.edu/projects/artmip/algorithms.html). The Sunway version CESM code is available at ZENODO via https://doi.org/10.5281/zenodo.3637771.

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

## Acknowledgements

This research is supported by the Science and Technology Innovation Program of Laoshan Laboratory (LSKJ202202503 to X.M.), Shandong Provincial Natural Science Foundation (ZR2022YQ29 to X.M.), Taishan Scholar Funds (tsqn202103028 to X.M.) and National Natural Science Foundation of China (41975065 to S.W.). We thank Sunway TaihuLight High-Performance Computer (Wuxi) and Laoshan Laboratory in Qingdao for providing the high resolution CESM simulations and high performance computing resources that contributed to the research results reported in this paper.

## Author contributions

S.W. performed most of the analyses under X.M.'s instruction. X.M. conceived the central ideal and wrote the manuscript. S.Z. assisted with the data processing and analyses. L.W. supervised the project. H.W. conducted the CESM simulations. Z.T. and G.X. assisted with storm track analysis and AR detection. Z.J., Z.C., and B.G. contributed to interpreting the results and improving the manuscript.

## Competing interests

The authors declare no competing interests.
