## [Peer Review File · Nature Communications]

Extreme Atmospheric Rivers in a Warming ClimateReviewers' comments:

Reviewer #1 (Remarks to the Author):

Review of Wang et al. 2021

Summary:

The authors investigate changes in the occurrence and characteristics of extreme atmospheric rivers (EARs) in a climate change context using a high-resolution, eddy-resolving global model (CESM). The authors report widespread & large increases in the frequency, intensity, and precipitation associated with EARs in key global AR regions under a high emissions/high warming future scenario. The vast majority of these increases appear to stem from thermodynamic changes to the AR environment, rather than dynamic ones. The authors further report that these projected increases in EARs are substantially larger in the high-resolution version of CESM vs. the coarse resolution version—with the implication that traditional GCM projections may underestimate climate change driven increases in atmospheric river frequency and intensity.

Overall assessment:

This is a very interesting analysis, and the underlying experiments are well-designed. The overall study findings have significant implications both within the atmospheric/climate science fields and from a societal perspective—it traditional GCM simulations really are underestimating plausible future increases in EAR activity, that would have serious consequences for climate adaptation activities and would motivate further study regarding the potential that other types of extreme events may be underestimated in present-day GCM simulations (including CMIP5, CMIP6, etc.) vs. eddy-resolving models.

For these reasons, I believe that this paper has strong potential to become a good candidate for publication in Nature Communications. However, I also believe this manuscript will require some major revisions before it reaches that point. I have enumerated specific points in the sections below.

Major comments:

1) To what extent can the authors really claim that the coarse-resolution simulations “underestimate” the EAR response to climate change? The high-resolution model does appear to have a high bias when it comes to the historical frequency/intensity of such events (though the coarse resolution has a low bias), per the supplementary info. Is it possible that the high-res simulations are overestimating the future increase in EARs, just as the low res simulations are potentially underestimating? Addressing this may not require new analysis, but I think it would be important to discuss this in more depth and perhaps to provide some additional quantitative context from the existing data (e.g. what is the ratio of the negative bias of the low res sims to the positive bias of the high-res sims? In other words, which is larger/more important)? This has important implications for everything from the paper’s title, key claims in the abstract, and elsewhere.

2) The present manuscript presents the finding that increases in extreme ARs are primarily driven by thermodynamic changes (vs dynamic changes) as novel, but there is no discussion that this has actually been reported by multiple other studies previously (e.g., Huang et al. 2020, Payne et al. 2020, Gao et al. 2015). The present manuscript may indeed be the first to point out that this fundamental driver is nearly ubiquitous in a global (vs. regional) context for EARs, but I still think the present discussion does lack contextualization among previous works in that regard. Please see my additional specific comments on this below.

3) While not seriously impeding overall understanding, there are multiple places where grammar, tense, and/or structural issues make specific sections or sentences difficult to parse or make what should be precise definitions a little ambiguous. The manuscript would strongly benefit from a close proof-reading to address these issues, but I think this would be easily achievable within the scope of a major revision.

Additional specific comments:

1) Figure 1 and associated discussion: The definition of extreme ARs used in this work seems to preclude them in some places where we know they do occur, such as west coast of NA (esp. California) and west coast of South America (esp. southern Chile). I suspect this is due to the inclusion of ARs from western boundary current (WBC) regions that have much higher background and extreme IVT than cool west coast continental regions (i.e., the coast of Japan vs. the coast of California). This is not a problem, per se, but does mean that the maps of EAR occurrence are curiously empty in some places where they are not only known to occur, but can be quite critical to flood risk and water supply (e.g., California, Chile, etc.). It might be worth considering whether a region-specific IVT threshold might be helpful in selecting EARs for the purpose of this work, so as to better represent events in non-WBC regions (though I don't consider that a hard requirement for publication—alternatively, discussion of these thresholds and their effects on the analysis could suffice).

2) The authors should note somewhere in the study that they use only the end-of-century RCP8.5 scenario as a point of comparison to historical conditions, which is a high emissions/high warming scenario. This is a perfectly acceptable methodological choice, but it would be worth either a) emphasizing that lesser warming scenarios would probably lead to a smaller increases in EARs, or b) actually providing some quantitative numbers for a mid-century period during RCP8.5 (say, 2030-2060 vs historical) if the data are readily available, as that may serve as a simple proxy for a "lower warming" scenario.

3) Lines 191-225: This section needs additional discussion/contextualization based on the existing literature. Multiple previous analyses have found that increases in AR (and, specifically, extreme ARs) in a warming climate were overwhelmingly due to the thermodynamic/moisture effect, with regionally-varying (but usually much smaller in magnitude, and more uncertain) contributions by dynamical changes. For example: Huang et al. 2020 found, using high-resolution model simulations nested within CESM-LENS, that increases in the intensity/precip associated with the most extreme California ARs in RCP8.5 were primarily (80%) due to thermodynamic changes, and only 20% attributable to dynamical changes. This is similar to earlier findings by Gao et al. 2015, which were further emphasized in the comprehensive review by Payne et al. 2020.

4) Line 52: Low bias has ambiguous meaning here (could be interpreted as "not having much bias." Instead, I would suggest "systematic negative intensity/frequency biases" or something similar.

References:

Huang et al. 2020:

Huang, X., Swain, D. L., & Hall, A. D. (2020). Future precipitation increase from very high resolution ensemble downscaling of extreme atmospheric river storms in California. *Science Advances*, 6(29), eaba1323. <http://advances.sciencemag.org/content/6/29/eaba1323.abstract>

Payne et al. 2020:

Payne, A. E., Demory, M.-E., Leung, L. R., Ramos, A. M., Shields, C. A., Rutz, J. J., et al. (2020). Responses and impacts of atmospheric rivers to climate change. *Nature Reviews Earth & Environment*, 1(3), 143-157. <https://doi.org/10.1038/s43017-020-0030-5>

Gao et al. 2015:

Gao, Y., Lu, J., Leung, L. R., Yang, Q., Hagos, S., & Qian, Y. (2015). Dynamical and thermodynamical modulations on future changes of landfalling atmospheric rivers over western North America. *Geophysical Research Letters*, 42(17), 7179-7186. <https://doi.org/10.1002/2015GL065435>. <https://doi.org/10.1002/2015GL065435>

Reviewer #2 (Remarks to the Author):

Reviewer's summary of manuscript

In "Severely Underestimated Extreme Atmospheric Rivers in a Warming Climate," Wang et al. analyze changes in extreme atmospheric rivers in a 150 year simulation with a high-res, coupled version of CESM (0.25-degree atmosphere, 0.1-degree ocean). They compare their results with a low-res configuration of CESM and with ERA5, and they argue that the higher resolution simulation has a much more realistic simulation of extreme IVT ($IVT > 1,250 \text{ kg/m/s}$) and its associated rainfall. They then analyze changes in extreme ARs in the high-resolution simulation (1956-2005 and 2050-2100) and show that the future simulations have large changes in the frequency of extreme ARs and associated precipitation: much larger than the change associated with non-extreme ARs. They decompose the change in extreme IVT and total IVT into dynamic and thermodynamic components and show that the thermodynamic (increase in moisture) change dominates and is slightly offset by an apparent poleward shift of the storm tracks leading to a decrease. The authors also examine landfalling ARs in the coastal margins of the Pacific and Atlantic and show that changes in extreme ARs are larger in the Atlantic than the Pacific and that there tend to be more extreme ARs in the western parts of the basins than the eastern parts (and changes are larger in these areas).

The three key takeaway messages of the manuscript appear to be:

1. HR simulations are necessary to simulate extreme ARs
2. extreme AR frequency doubles in future climate simulations
3. Thermodynamic changes (CC scaling) cause the increase in extreme ARs

Summary of review

While the manuscript does present some novel results that could potentially be of interest to a broader community, the manuscript has a number of significant drawbacks that limit its likely impact: particularly related to the robustness of the results. The most major issue is that the current manuscript does not consider several major sources of uncertainty that, if ignored, could lead to other groups coming to different conclusions if this analysis were repeated with different experimental choices. These issues, along with specific recommendations, are detailed below in the *Major issues* section.

I do think these issues are addressable in principle, though the extent of new analyses and revisions that seem to be necessary would make me lean toward recommending 'reject and resubmit' if this were a discipline-specific journal. The fact that this has been submitted to Nature Communications, I must additionally consider whether these results will be of high impact to the field. Based on my assessment of the key points of the manuscript (indicated above), I anticipate that only the second of the three takeaways ("extreme AR frequency doubles in future climate simulations") has the potential to be high impact. My assessment of the impact of these points is detailed in the *Impact of manuscript* section.

Given that the issues related to impact appear to be fundamental to the manuscript--and therefore cannot be changed--I therefore recommend that this manuscript not be considered for publication in Nature Communications. That said, the results presented in this manuscript are definitely of interest to the atmospheric and climate science community, and I expect that the authors would have an easy time getting through peer review in a discipline-specific journal (e.g., J. Climate, JGR-Atmospheres, or Climate Dynamics) if and when they submit a revised version.

I would also like to add that I know from quite a lot of personal experience how emotionally difficult it can be to receive a critical review of one's own work. I find that it can often feel like a critical review is a personal attack, particularly when the language in a review is blunt in its criticism. Because of that, I want to convey that I hold the authors in the highest respect and that I think that this is fundamentally good and interesting work. I will look forward to seeing this work evolve into its final form, whatever journal that ends up being in.

Major issues

Model and ARDT uncertainty

The manuscript's main results do not consider uncertainty in any form, let alone two relevant and major sources of uncertainty that have been documented in the literature: (1) structural uncertainty associated with the choice of modeling system, and (2) structural uncertainty associated with how AR boundaries are determined.

Model uncertainty

Admittedly, a full treatment of the first source of uncertainty would involve significant effort that would be out of scope for this paper. But HighResMIP output has been publicly available for some time, and it would be relatively straightforward to obtain and analyze a small subset of the simulation output to bolster some of the manuscript's major points: in particular that high-resolution models have significantly higher IVT than low-resolution models. Analyzing other simulations within the author's analytical framework seems particularly important, given that their results seem to conflict with a recent result reported by Reid et al. (2021). In Reid et al.'s Figure 3c, they show that the tails of the IVT PDF from a broad range of (low-resolution) CMIP6 simulations match well with ERA5. This appears to conflict with the author's implication (line 79, lines 86-88) that extreme ARs are severely underestimated in low resolution models.

**Recommendation:* the author's should somehow pull in information from other, independent simulations to investigate whether their results--especially with respect to the benefit of high resolution--extend to other modeling systems. Otherwise, this is a very model-specific paper that will likely be of limited utility and interest to researchers who don't use CESM as their primary simulation platform.

Reid, K. J., T. A. O'Brien, A. D. King, and T. P. Lane, 2021: Extreme Water Vapor Transport During the March 2021 Sydney Floods in the Context of Climate Projections. *Geophys. Res. Lett.*, 48, <https://doi.org/10.1029/2021GL095335>.

ARDT Uncertainty

The authors appear to have overlooked a significant emerging topic in the literature related to atmospheric rivers: uncertainty associated with the identification of AR boundaries. See, for example, the numerous publications on the Atmospheric River Tracking Method Intercomparison Project (ARMTIP) publication page: <https://www.cgd.ucar.edu/projects/artmip/publications.html>

Several recent studies associated with ARTMIP have shown that uncertainty associated with atmospheric river detection tools (ARDTs) can be quite large. In particular, some of the most recent results from ARTMIP indicate that conclusions drawn from investigations like this one may differ depending on which ARDT is chosen. This study uses one particular ARDT (though which one they use isn't clear -- see the point below about that), so it is not clear whether the results are robust to choice of ARDT.

**Recommendation:* The authors should repeat their analysis with at least one other ARDT. They should also discuss and cite relevant ARTMIP literature in their discussion of uncertainty. There are multiple ARDTs available in the public domain that the authors can choose from:

* <https://doi.org/10.1175/JCLI-D-15-0655.1>

* <https://doi.org/10.1002/2015JD024257>, <https://doi.org/10.1175/JHM-D-17-0114.1>

* <https://doi.org/10.5194/gmd-13-6131-2020>

* <https://doi.org/10.1029/2020JD033421>

Other sources of uncertainty

The authors also do not appear to report uncertainty in their results: e.g., are the differing results between LR- and HR-CESM statistically significant, and are the differences between the historical and RCP simulations statistically significant? The authors also do not consider uncertainty in their choice of threshold for defining extreme ARs (1,250 kg/m/s).

Recommendations: (1) the authors should include quantitative uncertainty analysis (e.g. statistical significance) when they present and discuss differences between two simulations (e.g., historical vs RCP 8.5 and LR vs HR), and (2) the authors should perform a sensitivity test with their extreme AR threshold.

Possible conflation of model resolution dependence and ARDT resolution dependence

The authors assert that high-res CESM is superior to low-res CESM due to the use of 'eddy-resolving resolution'. While this claim seems reasonable, it is also possible that the resolution dependence that the authors report is due to the ARDT itself; Reid et al. (2020) demonstrate that this can be the case for at least one ARDT. The authors should perform an analysis that tests the alternate hypothesis that the differing results between LR- and HR-CESM come from the resolution-dependence of the ARDT that the authors use.

Recommendation: The authors should repeat their analysis on a 'low-resolution' version of the HR-CESM simulations, in which the resolution of the HR-CESM fields are reduced by the use of an appropriate regridding method (e.g., conservative remapping). The authors should also pay special attention to the order of operations, as Reid et al. (2020) show that that also can have a significant impact (e.g., whether IVT is calculated before or after regridding), and the authors should discuss this in the manuscript.

Reid, K. J., A. D. King, T. P. Lane, and E. Short, 2020: The Sensitivity of Atmospheric River Identification to Integrated Water Vapor Transport Threshold, Resolution, and Regridding Method. *J. Geophys. Res. Atmos.*, 125, 1–15, <https://doi.org/10.1029/2020JD032897>.

Choice of time period

The HR CESM data extends from 1850-2100, however only the period from 1956-2005 and 2050-2100 are under consideration. An explanation for the choice of time period would provide better insight on the study. In particular, IVT is likely increasing rapidly in the 2050-2100 time period, so it is unclear whether a 50-year average is representative of conditions during that time. This also leads to the authors underutilizing their data, since information from the 2005-2050 timeperiod is ignored.

Recommendation: The authors should discuss their choice of time periods and indicate whether their main results are sensitive to which time period is chosen. They should also consider employing a trend analysis in lieu of their 'climatological difference analysis' approach, as a trend analysis would be able to utilize more of the simulation output and could therefore reduce uncertainty associated with internal variability.

Focus area of study

In the first paragraph of the manuscript (lines 41-46), the authors appear to argue that global AR analyses are important, which leads this reviewer (and this reviewer's students who provided input on the review) to think that the manuscript is going to be globally-oriented. While the first set of results (Figures 1 and 2) are indeed global, the manuscript becomes distinctly regional later in the manuscript, with most of the focus on the northern hemisphere (e.g., Figure 3, 4, 5b, and 5d and the related discussion).

Recommendation: The authors should either re-frame their introductory paragraph to avoid making it appear that the manuscript has a global focus, or--preferably--the authors should broaden the later part of their manuscript to focus more generally on AR landfalling regions in both the northern and southern hemisphere.

Interpretation of figures

The authors' analytic choices lead to quantities in some of their figures that have values that are difficult to interpret. In Figure 1, for example, the authors show extreme AR IVT and extreme AR precipitation and extreme AR precipitation and extreme AR precipitation and extreme AR

precipitation and extreme AR precipitation and extreme AR precipitation and extreme AR precipitation and extreme AR precipitation, but the colorbars for IVT don't even exceed 80 kg/m/s and the colorbars for precipitation don't exceed 2.4 mm/day. These values themselves are notably not extreme, which presumably arises due to the choice that "AR IVT (precipitation) is computed as the accumulated IVT (precipitation) of extreme ARs divided by the total number of winter season records" (lines 93-95). How should a general audience interpret these values? And is this the best way to report extreme IVT and precipitation in a high-impact journal? Figures 1-3 have a similar problem.

Relatedly, Figure 4c uses a strange quantity 'Total IVT' (also referred to as cross-sectional IVT in the caption) that has unusual units: $1e7$ kg/s. Based on the units and the brief statement in the caption, it seems that the authors must have integrated across the ARs in space; but it isn't clear from the text that this was done, and it isn't clear from the text **why** this was done.

Recommendation: The authors should revise their analysis to use quantities that help with communicating the concept that the ARs in HR-CESM are truly extreme. For example, the authors could consider normalizing IVT by a measure of the number of extreme AR timesteps (rather than the total number of winter season records) so that the quantities reported in Figs 1-3 are more representative of conditions in the extreme ARs when they occur. The authors should also consider whether 'Total IVT' is the most useful quantity to report, and if so, they should expand their discussion of this quantity since it's one that even people the AR research field will necessarily be familiar with.

Impact

As stated in above, the three key takeaway messages of the manuscript appear to be:

1. HR simulations are necessary to simulate extreme ARs
2. extreme AR frequency doubles in future climate simulations
3. Thermodynamic changes (CC scaling) cause the increase in extreme ARs

Of these three apparent takeaways, the first and second are minor variations on themes that are becoming ubiquitous in the literature: that higher resolution climate models are better for extremes, and that thermodynamic changes dominate changes in hydrologic cycle extremes (see some select references below). That is not to say that the minor variation presented in this manuscript (the focus on extreme ARs in particular) are irrelevant; rather, it seems more appropriate to convey these incremental advances in a discipline-specific journal.

The second of the three takeaways ("extreme AR frequency doubles in future climate simulations") has the potential to be high impact, as I am not aware of any similar result having been presented in the literature. If the paper were revised in such a way that this statement is clearly robust with respect to the various important sources of uncertainty, the manuscript as a whole might be of higher impact. However, addressing this would likely involve substantially rewriting the paper, as it would likely require analysis of HighResMIP simulations; I can't see how the authors would be able to do this without essentially writing a new paper altogether.

References related to high resolution and extremes:

- * Wehner, M., K. A. Reed, D. Stone, W. D. Collins, and J. Bacmeister, 2015: Resolution dependence of future tropical cyclone projections of CAM5.1 in the US CLIVAR Hurricane Working Group idealized configurations. *J. Clim.*, 150212130600007, <https://doi.org/10.1175/JCLI-D-14-00311.1>.
- * Rauscher, S. A., T. A. O'Brien, C. Piani, E. Coppola, F. Giorgi, W. D. Collins, and P. M. Lawston, 2016: A multimodel intercomparison of resolution effects on precipitation: simulations and theory. *Clim. Dyn.*, 47, 2205–2218, <https://doi.org/10.1007/s00382-015-2959-5>.
- * Scher, S., R. J. Haarsma, H. de Vries, S. S. Drijfhout, and A. J. van Delden, 2017: Resolution dependence of extreme precipitation and deep convection over the Gulf Stream. *J. Adv. Model. Earth Syst.*, 9, 1186–1194, <https://doi.org/10.1002/2016MS000903>.

References related to thermodynamic scaling and hydrological extremes:

* Emori, S., 2005: Dynamic and thermodynamic changes in mean and extreme precipitation under changed climate. *Geophys. Res. Lett.*, 32, L17706, <https://doi.org/10.1029/2005GL023272>.

* Gao, Y., J. Lu, L. R. Leung, Q. Yang, S. Hagos, and Y. Qian, 2015: Dynamical and thermodynamical modulations on future changes of landfalling atmospheric rivers over western North America. *Geophys. Res. Lett.*, 42, 7179–7186, <https://doi.org/10.1002/2015GL065435>.

* Prein, A. F., and L. O. Mearns, 2021: U.S. Extreme Precipitation Weather Types Increased in Frequency During the 20th Century. *J. Geophys. Res. Atmos.*, 126, 1–18, <https://doi.org/10.1029/2020JD034287>.

Minor and/or Specific Feedback

* Why RCP8.5 - not other scenarios? what are the implications of this?

* which ARDT was used -- it wasn't clear whether it was group's own or an existing one. The reference to Gimeno et al. (line 268) seems like it's intended to reference the ARDT used, but Gimeno et al. is just a short reference. Later in the paragraph (line 274), Xu et al. is referenced, but it isn't clear whether that reference is meant to justify the aspect ratio and isoperimetric quotient ratio or whether it's meant to indicate the ARDT that was used.

* line 34: ARs are not necessarily always associated with ETCs (e.g., see Zhang and Ralph 2019, Zhang et al. 2021). Also note that this is a misstatement of Ralph et al.; in coming up with the AMS glossary definition, they intentionally declined to state whether or not ARs are always associated with ETCs (hence the language 'often associated with ETCs')

* "characteristic PDFs" (e.g., line 167) <-- what is this? is it just a histogram? Or is it "PDF of characteristics"? I haven't heard this term before, and I am nominally in the same field of study as the authors of this manuscript.

Zhang, Z., F. M. Ralph, and M. Zheng, 2019: The Relationship Between Extratropical Cyclone Strength and Atmospheric River Intensity and Position. *Geophys. Res. Lett.*, 46, 1814–1823, <https://doi.org/10.1029/2018GL079071>.

Zhang, Z., and F. M. Ralph, 2021: The Influence of Antecedent Atmospheric River Conditions on Extratropical Cyclogenesis. *Mon. Weather Rev.*, 149, 1337–1357, <https://doi.org/10.1175/MWR-D-20-0212.1>.

Reply to referee #1

First, we would like to thank the referee for the invaluable comments. We have carefully considered each of your comments (*listed as bold and Italic below*) and revised the manuscript accordingly. Below are point-to-point replies to your comments:

Summary:

The authors investigate changes in the occurrence and characteristics of extreme atmospheric rivers (EARs) in a climate change context using a high-resolution, eddy-resolving global model (CESM). The authors report widespread & large increases in the frequency, intensity, and precipitation associated with EARs in key global AR regions under a high emissions/high warming future scenario. The vast majority of these increases appear to stem from thermodynamic changes to the AR environment, rather than dynamic ones. The authors further report that these projected increases in EARs are substantially larger in the high-resolution version of CESM vs. the coarse resolution version—with the implication that traditional GCM projections may underestimate climate change driven increases in atmospheric river frequency and intensity.

Overall assessment:

This is a very interesting analysis, and the underlying experiments are well-designed. The overall study findings have significant implications both within the atmospheric/climate science fields and from a societal perspective—it traditional GCM simulations really are underestimating plausible future increases in EAR activity, that would have serious consequences for climate adaptation activities and

would motivate further study regarding the potential that other types of extreme events may be underestimated in present-day GCM simulations (including CMIP5, CMIP6, etc.) vs. eddy-resolving models.

For these reasons, I believe that this paper has strong potential to become a good candidate for publication in Nature Communications. However, I also believe this manuscript will require some major revisions before it reaches that point. I have enumerated specific points in the sections below.

Major comments:

1) To what extent can the authors really claim that the coarse-resolution simulations “underestimate” the EAR response to climate change? The high-resolution model does appear to have a high bias when it comes to the historical frequency/intensity of such events (though the coarse resolution has a low bias), per the supplementary info. Is it possible that the high-res simulations are overestimating the future increase in EARs, just as the low-res simulations are potentially underestimating? Addressing this may not require new analysis, but I think it would be important to discuss this in more depth and perhaps to provide some additional quantitative context from the existing data (e.g. what is the ratio of the negative bias of the low-res sims to the positive bias of the high-res sims? In other words, which is larger/more important)? This has important implications for everything from the paper’s title, key claims in the abstract, and elsewhere.

A quantitative comparison of simulated EARs' bias between the high- and low-resolution models was provided in the revised manuscript. Also, following the 2nd reviewer's comments, the comparison was extended to a broader variety of climate models including low-resolution (LR) CMIP6 ($\sim 1^\circ$ atmosphere and ocean) and HighResMIP (see Methods for details). It needs to note that although currently available HighResMIP simulations that provide high frequency (6-hourly/daily) outputs to detect AR have "eddy-resolving" atmospheric resolution (0.25°), none of them are "eddy-resolving" in the ocean component (the finest ocean resolution is 0.25°). Compared with the ERA5 reanalysis, LR CMIP6 simulations underestimate the EARs by over 50% (revised **Fig.1f**) and the underestimates of EARs remain as high as $\sim 40\%$ in HighResMIP (**Fig. S2**). In contrast, the simulated EARs in HR CESM are significantly improved, although with slight overestimation ($\sim 10\%$) (**Fig.1f**). The results demonstrate that compared to "non-eddy-resolving" climate models, the "eddy-resolving" HR CESM does have improved capability in representing EARs. Quantified EAR biases in LR CMIP6, HighResMIP and HR CESM simulations were shown in revised Fig.1 and described in the text (See Lines **54-59**, **68-70**, **78-82**). Also, a quantitative description was included in the abstract (Lines **16,20**). The title was also revised to "*Extreme Atmospheric Rivers in a Warming Climate*".

Due to the slight overestimates of EARs in historical simulations in HR CESM, it is possible that projected EARs may be somewhat overestimated and more "eddy-resolving" climate models are required to evaluate the uncertainties in the future. We also note that ambiguities may exist in ERA5, bringing uncertainties to the model

validation. As reported in a recent study (Cobb et al. 2021), ERA5 IVT itself may be biased compared to drop-sonde observations. These points were now discussed in the revision (Lines 212-217): “There are still uncertainties that cannot be excluded in EAR projections in the study. It is possible the increase of EARs projected by HR CESM may be overestimated to some extent, and more “eddy-resolving” climate models are needed to evaluate the uncertainties in the future. Also, ERA5 IVT itself may be biased compared to dropsonde observations as reported in a recent study³³, which requires more drop-sonde observations at a global scale to reduce model validation uncertainties.”

2) The present manuscript presents the finding that increases in extreme ARs are primarily driven by thermodynamic changes (vs dynamic changes) as novel, but there is no discussion that this has actually been reported by multiple other studies previously (e.g., Huang et al. 2020, Payne et al. 2020, Gao et al. 2015). The present manuscript may indeed be the first to point out that this fundamental driver is nearly ubiquitous in a global (vs. regional) context for EARs, but I still think the present discussion does lack contextualization among previous works in that regard. Please see my additional specific comments on this below.

We agree and your criticism is well taken. Considering both your and the 2nd Reviewer’s concerns about the novelty of this point, the conclusion about the thermodynamic and dynamic contributions to EAR changes was removed from the major findings (see revised abstract), the related figure is moved to the Supplementary (Fig. S5), the corresponding section was cut down (Lines 130-142) and revised to

include detailed discussions on previous studies and to clarify the updates of our study (Lines 138-142): “The predominant role of thermodynamic response in determining AR projection under warming has been widely discussed in many previous studies at regional scales^{11,19,29} and the possible negative role of dynamic response in affecting AR changes has also been noted in the North Pacific and Mediterranean^{28,30}. It is further proved by the HR CESM that similar mechanisms can be applied at a global scale.”

In the revised manuscript, a new section discussing the changing relationship between ARs and storms was added. Generally, a reduced coupling between EARs and storms in a warming climate is found. This may have great implications for EAR prediction as the high internal variability induced by synoptic storms is the primary source that undermines the predictability of EARs. The key findings of the new section in the revised manuscript is: “*There will be a reduced coupling between EARs and storms in a warming climate, potentially influencing the predictability of future EARs.*” See more details of the new section in the revised manuscript (Lines 170-198).

3) While not seriously impeding overall understanding, there are multiple places where grammar, tense, and/or structural issues make specific sections or sentences difficult to parse or make what should be precise definitions a little ambiguous. The manuscript would strongly benefit from a close proof-reading to address these issues, but I think this would be easily achievable within the scope of a major revision.

We apologize for this and a careful proof-reading was made.

Additional Specific comments:

1) Figure 1 and associated discussion: The definition of extreme ARs used in this work seems to preclude them in some places where we know they do occur, such as west coast of NA (esp. California) and west coast of South America (esp. southern Chile). I suspect this is due to the inclusion of ARs from western boundary current (WBC) regions that have much higher background and extreme IVT than cool west coast continental regions (i.e., the coast of Japan vs. the coast of California). This is not a problem, per se, but does mean that the maps of EAR occurrence are curiously empty in some places where they are not only known to occur, but can be quite critical to flood risk and water supply (e.g., California, Chile, etc.). It might be worth considering whether a region-specific IVT threshold might be helpful in selecting EARs for the purpose of this work, so as to better represent events in non-WBC regions (though I don't consider that a hard requirement for publication—alternatively, discussion of these thresholds and their effects on the analysis could suffice).

Yes, the relatively indiscernible EAR IVT along the coastal regions (e.g., the west coast of North America) is because of the inclusion of much higher EAR IVT in the WBC regions. We want to clarify that Fig. 1 shows the accumulated EAR IVT which takes account of both the frequency and intensity of EARs. Separate plots of occurrence frequency and intensity of EARs were now shown in the supplementary (**Fig. S1**) and **Fig. R1** below. Taking the North Pacific in HR CESM as an example, the occurrence frequency of EARs along the west coast of North America is 2-5% while the occurrence

frequency is ~10% in the Kuroshio extension region (**Fig. R1b,d,f**). Also, the mean IVT intensity of EARs along the west coast of North America is ~600-800 kg/m/s and the value is ~1000 kg/m/s in the Kuroshio regions (**Fig. R1a,c,e**). The substantially lower accumulated EAR IVT along coastal regions shown in Fig. 1 is due to the combined impact of less frequent occurrence and weaker IVT intensity with a greater contribution from the former (**Fig. R1**), which is likely induced by the lower background IVT over the land than the WBC regions as you pointed out. To investigate the high-risk EARs in the coastal areas, we perform a separate analysis of landfalling EARs along the coastal regions in the later section (**Fig. 4**). A note was added to explain the indiscernible EAR IVT along the coasts in the revised manuscript (Lines **63-67**): “Note that EAR IVTs along the coasts are less evident than that in the western boundary current regions, due to lower occurrence frequency and weaker intensity of EARs (Fig. S1) which may be related to the lower background IVT over land than the warm ocean. Detailed analyses of high-risk landfalling EARs along the coasts are shown in a later section.”

We believe EARs along the coasts will become more apparent if using regional-specific IVT thresholds to define EARs. However, this might induce unfair comparisons of EARs among different regions for the following reasons: 1) EARs in the coastal regions will then be referenced to a lower IVT threshold and will be less hazardous. 2) Due to the large regional disparity of EARs, different thresholds may be required in different regions (e.g., the averaged EAR IVT along the coast of California is higher than that along the coast of Chile, Fig. R1a,e). To avoid these, a ubiquitous

IVT threshold is applied throughout the globe in the manuscript. This does obscure weaker EARs when mapping all EARs together, but it allows a fair comparison of global EARs among different basins. This was now explained in Methods (Lines 283-287): “Considering the large regional disparity of EARs, the application of regional-specific IVT thresholds to define EARs may help to well represent AR features in specific regions. However, it will also induce unfair comparisons of EARs among different regions. To avoid this, a ubiquitous IVT threshold is applied throughout the globe in this study, which allows a fair comparison of global EARs among different basins.”

Fig. R1. The mean IVT intensity and occurrence frequency of EARs in HR CESM. The mean IVT intensity (**c**, kg/m/s) and occurrence frequency (**d**, %) simulated in HR-HIS (1956-2005). **a&e**, **b&f** are the zoomed-in plots along the coasts of California and Chile.

2) The authors should note somewhere in the study that they use only the end-of-century RCP8.5 scenario as a point of comparison to historical conditions, which is a high emissions/high warming scenario. This is a perfectly acceptable methodological choice, but it would be worth either a) emphasizing that lesser

warming scenarios would probably lead to a smaller increases in EARs, or b) actually providing some quantitative numbers for a mid-century period during RCP8.5 (say, 2030-2060 vs historical) if the data are readily available, as that may serve as a simple proxy for a “lower warming” scenario.

Thanks for pointing this out. The sensitivity of EAR response to different warming levels is evaluated and added in the revision (revised **Fig. 3**). The result shows that the EARs increase almost linearly with temperature warming. The estimated increasing rate of EAR-related IVT during 2030-2100 is $\sim 70 \text{ kg/m/s/0.2}^\circ\text{C}$ per decade ($\sim 25\%$ per decade in reference to historical value). A lower-level warming period (2030-2050) indeed corresponds to a less intense increase of EARs. Specifically, the accumulated IVT of EARs during the mid-century is approximately 1.45 times the historical value but half the value of 2100 (high warming period). See detailed descriptions in the revised manuscript (**Fig. 3** and Lines **115-124**).

3) Lines 191-225: This section needs additional discussion/contextualization based on the existing literature. Multiple previous analyses have found that increases in AR (and, specifically, extreme ARs) in a warming climate were overwhelmingly due to the thermodynamic/moisture effect, with regionally-varying (but usually much smaller in magnitude, and more uncertain) contributions by dynamical changes. For example: Huang et al. 2020 found, using high-resolution model simulations nested within CESM-LENS, that increases in the intensity/precip associated with the most extreme California ARs in RCP8.5 were primarily (80%) due to thermodynamic changes, and only 20% attributable to dynamical changes. This is similar to earlier

findings by Gao et al. 2015, which were further emphasized in the comprehensive review by Payne et al. 2020.

As replied to your overall comment 2), your criticism is well taken. The conclusion about the thermodynamic and dynamic contributions to EAR changes was removed from the major findings, the related figure is moved to the Supplementary (**Fig. S5**) and the corresponding section was cut down and revised to include detailed discussions on previous studies (Lines **130-142**). More details can be found in the revised manuscript and in replies to your major comments 2).

4) Line 52: Low bias has ambiguous meaning here (could be interpreted as "not having much bias." Instead, I would suggest "systematic negative intensity/frequency biases" or something similar.

Revised. Thanks. See Line **52**.

References

1. Lavers, D. A. *et al.* Future changes in atmospheric rivers and their implications for winter flooding in Britain. *Environ. Res. Lett.* **8**, 034010 (2013).
2. Payne, A. E. *et al.* Responses and impacts of atmospheric rivers to climate change. *Nat. Rev. Earth Environ.* **1**, 143–157 (2020).
3. Gao, Y. *et al.* Dynamical and thermodynamical modulations on future changes of landfalling atmospheric rivers over western North America. *Geophys. Res. Lett.* **42**, 7179–7186 (2015).
4. Huang, X., Hall, A. D. & Berg, N. Anthropogenic Warming Impacts on Today's Sierra Nevada Snowpack and Flood Risk. *Geophys. Res. Lett.* **45**, 6215–6222 (2018).
5. Polade, S. D., Gershunov, A., Cayan, D. R., Dettinger, M. D. & Pierce, D. W. Precipitation in a warming world: Assessing projected hydro-climate changes in California and other Mediterranean climate regions. *Sci Rep* **7**, 10783 (2017).
6. Cobb, A., Delle Monache, L., Cannon, F. & Ralph, F. M. Representation of Dropsonde-Observed Atmospheric River Conditions in Reanalyses. *Geophys. Res. Lett.* **48**, (2021).

Reply to referee #2

First, we would like to thank the referee for the invaluable comments and the detailed explanation of your concerns. After carefully considering each of your comments (*listed as bold and Italic below*), we made great efforts to address all your criticisms and kindly ask you to review the revised manuscript again. Below are point-to-point replies to your comments:

Reviewer's summary of manuscript

In "Severely Underestimated Extreme Atmospheric Rivers in a Warming Climate," Wang et al. analyze changes in extreme atmospheric rivers in a 150 year simulation with a high-res, coupled version of CESM (0.25-degree atmosphere, 0.1-degree ocean). They compare their results with a low-res configuration of CESM and with ERA5, and they argue that the higher resolution simulation has a much more realistic simulation of extreme IVT ($IVT > 1,250 \text{ kg/m/s}$) and its associated rainfall. They then analyze changes in extreme ARs in the high-resolution simulation (1956-2005 and 2050-2100) and show that the future simulations have large changes in the frequency of extreme ARs and associated precipitation: much larger than the change associated with non-extreme ARs. They decompose the change in extreme IVT and total IVT into dynamic and thermodynamic components and show that the thermodynamic (increase in moisture) change dominates and is slightly offset by an apparent poleward shift of the storm tracks leading to a decrease. The authors also examine landfalling ARs in the coastal margins of the Pacific and Atlantic and show that changes in extreme ARs are larger in the Atlantic than the Pacific and that there

tend to be more extreme ARs in the western parts of the basins than the eastern parts (and changes are larger in these areas).

The three key takeaway messages of the manuscript appear to be:

- 1. HR simulations are necessary to simulate extreme ARs*
- 2. extreme AR frequency doubles in future climate simulations*
- 3. Thermodynamic changes (CC scaling) cause the increase in extreme ARs*

Summary of review

*While the manuscript does present some novel results that could potentially be of interest to a broader community, the manuscript has a number of significant drawbacks that limit its likely impact: particularly related to the robustness of the results. The most major issue is that the current manuscript does not consider several major sources of uncertainty that, if ignored, could lead to other groups coming to different conclusions if this analysis were repeated with different experimental choices. These issues, along with specific recommendations, are detailed below in the *Major issues* section.*

I do think these issues are addressable in principle, though the extent of new analyses and revisions that seem to be necessary would make me lean toward recommending 'reject and resubmit' if this were a discipline-specific journal. The fact that this has been submitted to Nature Communications, I must additionally consider whether these results will be of high impact to the field. Based on my assessment of the key points of the manuscript (indicated above), I anticipate that only the second of the three takeaways ("extreme AR frequency doubles in future

*climate simulations") has the potential to be high impact. My assessment of the impact of these points is detailed in the *Impact of manuscript* section.*

Given that the issues related to impact appear to be fundamental to the manuscript--and therefore cannot be changed--I therefore recommend that this manuscript not be considered for publication in Nature Communications. That said, the results presented in this manuscript are definitely of interest to the atmospheric and climate science community, and I expect that the authors would have an easy time getting through peer review in a discipline-specific journal (e.g., J. Climate, JGR-Atmospheres, or Climate Dynamics) if and when they submit a revised version.

I would also like to add that I know from quite a lot of personal experience how emotionally difficult it can be to receive a critical review of one's own work. I find that it can often feel like a critical review is a personal attack, particularly when the language in a review is blunt in its criticism. Because of that, I want to convey that I hold the authors in the highest respect and that I think that this is fundamentally good and interesting work. I will look forward to seeing this work evolve into its final form, whatever journal that ends up being in.

Many thanks for your constructive comments and we appreciate your detailed explanation. Careful considerations are taken to address your two major concerns in the revised manuscript:

1) **Result uncertainties.** A comprehensive evaluation with regard to model uncertainties, ARDT uncertainties and other possible sources of uncertainties (data resolution, IVT thresholds to define EARs, different warming levels and significance

test) was conducted following your suggestions. The results verified the severe underestimates of EARs in current “non-eddy-resolving” climate models (including LR CMIP6 and HighResMIP) and the substantial improvement in HR CESM. It is also well evidenced that the projected EAR changes in HR CESM (“*a global doubling of EARs (>1250 kg/m/s) by the end of the 21st century*”) is robust. See details in replies to your comments on “major issues” below.

2) Impact of the 1st and 3rd key points.

For the 1st key point, EAR analyses were extended to a broad variety of climate models including LR CMIP6 and HighResMIP. Ubiquitous underestimates of EARs are observed in these “non-eddy-resolving” climate models. Although it has been well recognized that increased model resolution will benefit the simulation of extremes, long-term “eddy-resolving” climate simulations for both the ocean and the atmosphere were not available before HR CESM was reported in the manuscript. Even in HighResMIP, the underestimates of EARs remain as high as ~40%, likely due to the restricted eddy-resolving capability of the ocean component. With significantly improved EAR simulations, HR CESM provides a unique opportunity to examine the response of EARs to anthropogenic warming that may be severely biased in previous climate simulations. See details in replies to your comments on “impact” below.

The 1st key point we want to address in the revised manuscript is: “*EARs are severely underestimated in current ‘non-eddy-resolving’ climate models, casting significant uncertainties on their projections. This problem is largely solved in the ‘eddy-resolving’ HR CESM.*” To the best of our knowledge, this is the first time that

such a severe bias of EAR simulation in current climate models is reported and quantified, and a substantial improvement in “eddy-resolving” coupled climate simulations is shown at a global scale, which possibly leads to significant improvement on EAR projections.

For the 3rd key point, the conclusion about the thermodynamic and dynamic contributions to EAR changes under warming was removed from the major findings. A new section discussing the changing relationship between ARs and storms was added. Generally, a reduced coupling between EARs and storms in a warming climate is found. This may have great implications for EAR prediction as the high internal variability induced by synoptic storms is the primary source that undermines the predictability of EARs. The 3rd key point we want to address in the revised manuscript is: “*The coupling relationship between EARs and storms will be reduced in a warming climate, potentially influencing the predictability of future EARs.*” See more details in replies to your comments on “impact” below.

In summary, new datasets and new analyses were included and are supportive of the major findings. The result's credibility is also confirmed by a comprehensive evaluation of possible uncertainties. We believe all your concerns are well addressed in the revised manuscript. We further believe these revised findings are novel, robust and of great interest to the broad communities in climate science, and are worthwhile to be reported in *Nature Communications*.

Major issues

Model and ARDT uncertainty

The manuscript's main results do not consider uncertainty in any form, let alone two relevant and major sources of uncertainty that have been documented in the literature: (1) structural uncertainty associated with the choice of modeling system, and (2) structural uncertainty associated with how AR boundaries are determined.

Model uncertainty

Admittedly, a full treatment of the first source of uncertainty would involve significant effort that would be out of scope for this paper. But HighResMIP output has been publicly available for some time, and it would be relatively straightforward to obtain and analyze a small subset of the simulation output to bolster some of the manuscript's major points: in particular that high-resolution models have significantly higher IVT than low-resolution models. Analyzing other simulations within the author's analytical framework seems particularly important, given that their results seem to conflict with a recent result reported by Reid et al. (2021). In Reid et al.'s Figure 3c, they show that the tails of the IVT PDF from a broad range of (low-resolution) CMIP6 simulations match well with ERA5. This appears to conflict with the author's implication (line 79, lines 86-88) that extreme ARs are severely underestimated in low resolution models.

**Recommendation:* the author's should somehow pull in information from other, independent simulations to investigate whether their results--especially with respect to the benefit of high resolution--extend to other modeling systems. Otherwise, this is a very model-specific paper that will likely be of limited utility and interest to researchers who don't use CESM as their primary simulation platform.*

Following your suggestion, more model results from HighResMIP and LR CMIP6 were analyzed. Only four HighResMIP simulations were found that provide high frequency (6-hourly/daily) IVT or three-dimensional u, v, q, which are needed to detect ARs. One of these HighResMIP simulations has a comparable atmospheric resolution (0.25°) to HR CESM, but none of them are “eddy-resolving” (0.1°) in the ocean component (the finest ocean resolution is 0.25°). For comparison, four paring LR ($\sim 1^\circ$ atmosphere and ocean) CMIP6 simulations were selected (see Methods for details). Compared to ERA5, LR CMIP6 underestimates the EARs by over 50% (**Fig. R2a,c,e** below and revised **Fig.1**). Enhanced EARs are simulated in HighResMIP with increased resolution, but the amplitude is still much lower than ERA5 (**Fig. R2d** and **Fig. S2**). The underestimates of EARs in HighResMIP remain as high as $\sim 40\%$ (**Fig. R2e** and **Fig. S2**), likely due to the restricted eddy-resolving capability of the ocean component. In contrast, EAR simulations in “eddy-resolving” HR CESM are substantially improved (despite a slight overestimate of $\sim 10\%$), comparable to ERA5 (**Fig. R2b,e** and revised **Fig.1**). The results suggest that the current “non-eddy-resolving” climate models severely underestimate EARs, “eddy-resolving” climate models for both the atmosphere and the ocean are required to get more realistic EAR simulations.

The projected EAR changes in CMIP6 (including LR CMIP6 and HighResMIP) are also evaluated and compared with HR CESM in the overlapping periods (2030-2050, revised **Fig. 3b,c**). Although increased EARs in a warming period are projected by all models, the estimated EAR occurrence frequency and accumulated IVT in RCP

are only half of HR CESM values, indicating that EAR projections based on “non-eddy-resolving” climate simulations may also be severely underestimated.

Together, the extended analyses of LR CMIP6 and HighResMIP confirm the poor ability of current climate models in simulating and projecting EARs. On the other hand, the “eddy-resolving” HR CESM provides a unique tool to examine EARs changes under global warming with higher confidence. In the revised manuscript, simulated and projected EAR results from LR CMIP6 and HighResMIP were added and compared with HR CESM (revised **Figs. 1, 3** and **Fig. S2**), which confirmed the severe negative EAR simulation biases in current climate models and the substantial improvement of HR CESM (Lines 44-95).

Figure R2 | Observed and simulated EARs. Normalized accumulated EAR IVT (kg/m/s) in ERA5 reanalysis (a), HR-CESM (b), LR-CMIP6 (c) and HighResMIP (d) during 1979-2005. The global averaged EAR IVT (kg/m/s) for ERA5 reanalysis, HR-CESM, LR-CMIP6 and HighResMIP during 1979-2005 (e).

Thank you for bringing up the recent study by Reid et al. (2021) to us. After carefully reading their paper, we found considerable differences between their paper and our study: 1) Only a small region around Sydney was analyzed in their paper while a global analysis was performed in our study; 2) The maximum IVT is cut off at 800 kg/m/s in their study while we focused on EARs that exceeds 1250 kg/m/s; 3) A logarithm instead of original pdf of IVT is plotted in their study. We repeated their analysis using LR CMIP6 selected in our study, the comparison of logarithm IVT pdf in Sydney between LR CMIP6 and ERA5 is shown below (**Fig. R3a**). The plot is highly consistent with Reid's result (**Fig. R3b**): LR CMIP6 reproduces the IVT pdf reasonably well although we can see differences begin to emerge at the high-end tail. The results suggest that LR CMIP may well represent IVT distribution in specific regions, especially for moderate IVT strength. However, on a global scale, the primarily hazardous EARs ($IVT > 1250 \text{ kg/m/s}$) are severely underestimated as shown in Fig.1 in the manuscript.

Figure R3 | (a) Semi-log probability distribution of mean IVT at 34°S in ERA5 (orange) and LR-CMIP6 (blue) during 1979-2005. **b**, same plot from Reid et al.'s Figure 3c.

ARDT Uncertainty

The authors appear to have overlooked a significant emerging topic in the literature related to atmospheric rivers: uncertainty associated with the identification of AR boundaries. See, for example, the numerous publications on the Atmospheric River Tracking Method Intercomparison Project (ARTMIP) publication page: <https://www.cgd.ucar.edu/projects/artmip/publications.html>

Several recent studies associated with ARTMIP have shown that uncertainty associated with atmospheric river detection tools (ARDTs) can be quite large. In particular, some of the most recent results from ARTMIP indicate that conclusions drawn from investigations like this one may differ depending on which ARDT is chosen. This study uses one particular ARDT (though which one they use isn't clear -- see the point below about that), so it is not clear whether the results are robust to choice of ARDT.

**Recommendation:* The authors should repeat their analysis with at least one other ARDT. They should also discuss and cite relevant ARTMIP literature in their discussion of uncertainty. There are multiple ARDTs available in the public domain that the authors can choose from:*

** <https://doi.org/10.1175/JCLI-D-15-0655.1>*

** <https://doi.org/10.1002/2015JD024257>, <https://doi.org/10.1175/JHM-D-17-0114.1>*

** <https://doi.org/10.5194/gmd-13-6131-2020>*

* <https://doi.org/10.1029/2020JD033421>

Many thanks for your constructive comments and we completely agree that the application of different ARDTs is necessary to evaluate the result uncertainties. Except for the fixed IVT250 method used in the previous manuscript, two additional ARDTs from ARTMIP--IVT85% (spatial-varying IVT threshold, Guan and Waliser et al. 2015) and IPART (threshold-free IVT based on image-processing, Xu et al. 2020) are applied in HR CESM to verify the robustness of simulated and projected EARs in HR CESM.

Figure R4 below shows the accumulated EAR IVT simulated in HR-HIS and the project changes in HR-RCP based on IVT85% and IPART ARDTs, respectively. Compared to IVT250 results shown in Fig.2 in the manuscript, although the amplitude of accumulated EAR IVT becomes higher (primarily due to greater numbers of EARs detected) for IVT85% and IPART methods, the global distributions of EARs are remarkably similar across all chosen ARDTs (**Fig. R4a,d**). Moreover, the projected EAR changes under global warming are also highly consistent, all demonstrating a global doubling of EARs again with similar spatial distribution in future climate(**Fig. R4b,c,e,f**).

The above results suggest that uncertainties induced by ARDTs mainly influence the absolute amplitude of EAR IVT, which modifies the historical and future simulations equally. However, the projected EARs changes are less influenced, indicating that “the global doubling of EARs under warming in HR CESM” is robust. Result sensitivities to ARDT uncertainties were added in the supplementary (**Fig. S4**) and related discussions were updated in the revision(**Lines 109-114**): “Existing studies have noted that considerable uncertainties may be induced in AR statistics by different

AR detection tools (ARDTs)²⁷. Two additional ARDTs (Methods) are applied to verify the robustness of EAR projections in HR CESM. Although the amplitude of detected EARs varies among different ARDTs (Fig. S4), the projected EAR changes under global warming are highly consistent, all demonstrating a global doubling of EARs with similar spatial distribution in future climate.”

Figure R4 | Sensitivity of EAR projections to ARDTs in HR CESM. a, b, as for Fig. 2d, e, but for IPART method. (c) Global averaged EAR IVT (kg/m/s) in HR-HIS and HR-RCP based on IPART method. **d-f,** as for a-c, but for IVT85% ARDT.

Other sources of uncertainty

The authors also do not appear to report uncertainty in their results: e.g., are the differing results between LR- and HR-CESM statistically significant, and are the differences between the historical and RCP simulations statistically significant? The authors also do not consider uncertainty in their choice of threshold for defining extreme ARs (1,250 kg/m/s).

Recommendations: (1) *the authors should include quantitative uncertainty*

**analysis (e.g statistical significance) when they present and discuss *differences*

*between two simulations (e.g., historical vs RCP 8.5 and LR vs *HR), and (2) the authors should perform a sensitivity test with their extreme *AR threshold.*

Many thanks for your comments and these points are well taken. Two-sided Student's t-tests were applied when computing differences between related simulations and differences significant above 95% confidence level are shaded by gray dots (See revised **Figs. 2, 4, 5** and **Figs. S3-S5, S7**).

To test result sensitivities to different IVT thresholds used to define EARs, we repeated the analysis by redefining EARs using the 75th and 90th percentile values of AR IVT derived from the entire historical simulations of global ARs, which corresponds to 1581 kg/m/s and 1336 kg/m/s IVT threshold, respectively. The results (**Fig. R5**) show that simulated and projected EARs defined by 75th percentile threshold have similar amplitude and distribution, compared to that shown in Fig. 2 in both HR-HIS and HR-RCP, with an approximate twofold global increase under warming (**Fig. R5d-f**). The simulated EAR IVT in HR-HIS defined by 90th percentile threshold is much lower than that shown in Fig. 2 because a lower number of EARs are above the 90th percentile IVT threshold (**Fig. R5a**). However, the amplification of EARs in HR-RCP is even higher (~3 times the historical value, **Fig. R5b,c**). The analyses suggest that higher thresholds tend to result in fewer EARs but more intense amplifications of EARs' response to global warming. There is no uniform definition of EARs, the reason we choose 1250 kg/m/s threshold is that these EARs are claimed to be primarily hazardous according to a recent study that evaluated the impact of different category ARs (Ralph et al. 2019). The sensitivity test to different IVT thresholds was included

in the supplementary (**Fig. S7**) and a discussion on this was added in the revision (**Lines 217-223**): “Moreover, it needs to point out that the primarily hazardous EARs (IVT >1250 kg/m/s) are chosen in this study according to a recent classification²⁰ and the amplitude of EAR projections may vary with different EAR definitions. Sensitivity tests of EARs defined by different IVT thresholds (the 75th and 90th percentile values, **Fig. S7**) suggest that a higher IVT threshold is likely to generate fewer EARs but more intense EARs’ increase to global warming.”

Fig. R5 | Sensitivity of EAR projections to different IVT thresholds used to define EARs in HR CESM. a, b, as for **Fig. 2d, e**, but for EARs defined by the 90th percentile IVT threshold. **(c)** Global averaged EAR IVT (kg/m/s) in HR-HIS and HR-RCP for EARs defined by the 90th percentile IVT threshold. **d-f**, as for **a-c**, but for EARs defined by the 75th percentile IVT threshold.

Possible conflation of model resolution dependence and ARDT resolution dependence

The authors assert that high-res CESM is superior to low-res CESM due to the use of 'eddy-resolving resolution'. While this claim seems reasonable, it is also possible that the resolution dependence that the authors report is due to the ARDT itself; Reid et al. (2020) demonstrate that this can be the case for at least one ARDT.

The authors should perform an analysis that tests the alternate hypothesis that the differing results between LR- and HR-CESM come from the resolution-dependence of the ARDT that the authors use.

**Recommendation:* The authors should repeat their analysis on a 'low-resolution' version of the HR-CESM simulations, in which the the resolution of the HR-CESM fields are reduced by the use of an appropriate regridding method (e.g., conservative remapping). The authors should also pay special attention to the order of operations, as Reid et al. (2020) show that that also can have a significant impact (e.g., whether IVT is calculated before or after regridding), and the authors should discuss this in the manuscript.*

Following your suggestions, the 0.25° IVT in HR CESM is regridded onto 1° LR grid and ARs are detected using the regridded LR IVT (due to the lack of three-dimensional u, v, q outputs in CESM, we cannot do the regridding before IVT is calculated). The EAR differences between regridded HR and LR CESM (**Fig. R6b**) are remarkably comparable to that computed on original HR grids (**Fig. R6a**), although a close look at the global averaged EARs shows a slight decrease of EAR amplitude on the regridded grids (**Fig. R6c**). The results indicate that the substantially improved simulation of EARs in HR CESM should not be much influenced by data resolution used in AR detection. In fact, further analyses demonstrate that the improvements of EARs in HR CESM are attributed to modifications of both thermodynamic and dynamic fields (See **Fig. S3** and detailed discussions in the manuscript). The above points were clarified in the revision (Lines **82-87**): “The improvement of EARs is

independent of the resolution of IVT data as EARs of similar amplitude are observed when regridding the HR IVT onto the LR grid. Further comparison of simulated EARs between HR and a set of parallel LR CESM simulations (Methods) indicates that the better-resolved EARs in HR CESM are resulted from both thermodynamic and dynamic improvements with a higher contribution from the latter (**Fig. S3**).”

Figure R6. (a) The difference of EAR IVT (kg/m/s) between HR and LR CESM in historical simulations (1956-2005). (b), as for (a), but for the difference of that between regridded HR and LR CESM. (c) Global averaged EAR IVT (c, kg/m/s) in HR, LR and regridded HR CESM.

Choice of time period

The HR CESM data extends from 1850-2100, however only the period from 1956-2005 and 2050-2100 are under consideration. An explanation for the choice of time period would provide better insight on the study. In particular, IVT is likely increasing rapidly in the 2050-2100 time period, so it is unclear whether a 50-year average is representative of conditions during that time. This also leads to the authors underutilizing their data, since information from the 2005-2050 timeperiod is ignored.

**Recommendation:* The authors should discuss their choice of time periods and indicate whether their main results are sensitive to which time period is chosen. They should also consider employing a trend analysis in lieu of their 'climatological difference analysis' approach, as a trend analysis would be able to utilize more of the*

simulation output and could therefore reduce uncertainty associated with internal variability.

In HR CESM, RCP8.5 warming forcing is activated from 2006 onward. To allow a complete model adjustment to the switched forcing and to maximize the warming effect, a later period (2051-2100) in future simulations and a corresponding historical period (1956-2005) were chosen. This was now explained in Methods (Lines **233-238**).

Following your suggestions, the sensitivity of EAR response to different warming levels is evaluated in the revised manuscript (**Fig. R7** below and revised **Fig. 3**). The trend analysis shows that EARs increase almost linearly with temperature warming (**Fig. R7a**). The estimated increasing rate of EAR-related IVT during 2030-2100 is $\sim 70 \text{ kg/m/s/0.2}^\circ\text{C}$ per decade ($\sim 25\%$ per decade in reference to historical EAR IVT). In a lower-level warming scenario (2030-2050), a less intense EAR increase is found. The accumulated IVT of EARs during the mid-century is approximately 1.45 times the historical value (**Fig. R7c**) but half the value of 2100 (high warming period) (**Fig. R7a**). Analyses of EAR response to different warming levels and related descriptions were updated in the revised manuscript (See revised **Fig. 3** and detailed discussion in Lines **115-129**).

Figure R7 | EAR response to different warming levels. Time series of global averaged EAR IVT (blue line) and SST (red line) from 2030 to 2100 in HR CESM (a). The blue shading outlines the maximum and minimum EAR IVTs averaged in the NP, NA, SP, SA. Global averaged EAR occurrence frequency (b, %) and normalized accumulated IVT (c, kg/m/s) in historical (1980-2000) and future (2030-2050) periods in HR CESM and CMIP6 (including LR CMIP6 and HighResMIP).

Focus area of study

In the first paragraph of the manuscript (lines 41-46), the authors appear to argue that global AR analyses are important, which leads this reviewer (and this reviewer's students who provided input on the review) to think that the manuscript is going to be globally- oriented. While the first set of results (Figures 1 and 2) are indeed global, the manuscript becomes distinctly regional later in the manuscript, with most of the focus on the northern hemisphere (e.g., Figure 3, 4, 5b, and 5d and the related discussion).

**Recommendation:* The authors should either re-frame their introductory paragraph to avoid making it appear that the manuscript has a global focus, or--preferably--the authors should broaden the later part of their manuscript to focus*

more generally on AR landfalling regions in both the northern and southern hemisphere.

The study focuses on global EARs and we apologize for any misleading discussions in the previous manuscript. **Fig. 4** was revised to include landfalling EARs in both the northern and southern hemispheres (the number of EARs landfalling along the east coast of the South Atlantic is small and too weak to be shown). The section about landfalling EARs' response to global warming was reorganized and the related descriptions were largely rewritten to avoid regional focus, and highlight the consistent global responses as well as the disproportionate increase between landfalling EARs and total EARs in the revision (See details in Lines **143-169**). Besides, the new analysis of the relationship between EARs and storms (**Fig. 5**) also includes both northern and southern hemisphere EARs.

Interpretation of figures

The authors' analytic choices lead to quantities in some of their figures that have values that are difficult to interpret. In Figure 1, for example, the authors show extreme AR IVT and extreme AR precipitation, but the colorbars for IVT don't even exceed 80 kg/m/s and the colorbars for precipitation don't exceed 2.4 mm/day. These values themselves are notably not extreme, which presumably arises due to the choice that "AR IVT (precipitation) is computed as the accumulated IVT (precipitation) of extreme ARs divided by the total number of winter season records" (lines 93-95). How should a general audience interpret these values? And is this the best way to

report extreme IVT and precipitation in a high-impact journal? Figures 1-3 have a similar problem.

*Relatedly, Figure 4c uses a strange quantity 'Total IVT' (also referred to as cross-sectional IVT in the caption) that has unusual units: $1e7$ kg/s. Based on the units and the brief statement in the caption, it seems that the authors must have integrated across the ARs in space; but it isn't clear from the text that this was done, and it isn't clear from the text **why** this was done.*

****Recommendation:*** The authors should revise their analysis to use quantities that help with communicating the concept that the ARs in HR-CESM are truly extreme. For example, the authors could consider normalizing IVT by a measure of the number of extreme AR timesteps (rather than the total number of winter season records) so that the quantities reported in Figs 1-3 are more representative of conditions in the extreme ARs when they occur. The authors should also consider whether 'Total IVT' is the most useful quantity to report, and if so, they should expand their discussion of this quantity since it's one that even people the AR research field will necessarily be familiar with.*

We used accumulated IVT to get a full assessment of the combined impact of the frequency and intensity change of EARs, but as you pointed out, the division of accumulated IVT by the whole winter season records creates much lower IVT value than individual EARs. Following your advice, the accumulated IVT is now normalized by the climatological EAR occurrence number to improve the readability. To reserve the complete information of the frequency and intensity in the normalized IVT, a

ubiquitous EAR number derived from HR-HIS (the maximum occurrence frequency of climatological EARs is 10%, corresponding to 72 EARs per winter) is used at all grid points for all simulations. The normalization generates IVT values of ~800 kg/m.s, representative of the averaged EAR intensity (e.g., see revised Fig.1). In the revised manuscript, all related figures were replotted (**Figs. 1-4 and Figs. S2-S5, S7**) and the corresponding descriptions were updated. Calculation details of the accumulated IVT and the normalization were explained in Methods (Lines **288-296**): “Accumulated IVT is computed to get a full assessment of the combined impact of the frequency and intensity change of EARs. To improve the readability, the accumulated IVT is then normalized by a ubiquitous EAR occurrence number to represent the typical EAR strength. The ubiquitous EAR occurrence number is derived as the maximum occurrence of climatological EARs in HR-HIS, corresponding to (49) 72 (landfalling) EARs per winter. Specifically, the EAR IVT at each grid is computed as $\frac{\sum_{EAR=1}^n IVT}{N}$ (where IVT is the intensity of an individual EAR, n is the total number of EARs at the grid point, N is the climatological EAR occurrence number used in the normalization). A similar calculation is applied for precipitation.” Additionally, the occurrence frequency and intensity of EARs are shown separately in the supplementary (**Fig. S1**) for reference.

In the revision, the section on landfalling EARs response to global warming was reorganized and Fig. 4 was replotted. The “cross-sectional total IVT” panel was removed from the new Fig. 4.

Impact

As stated in above, the three key takeaway messages of the manuscript appear to be:

- 1. HR simulations are necessary to simulate extreme ARs*
- 2. extreme AR frequency doubles in future climate simulations*
- 3. Thermodynamic changes (CC scaling) cause the increase in extreme ARs*

Of these three apparent takeaways, the first and second are minor variations on themes that are becoming ubiquitous in the literature: that higher resolution climate models are better for extremes, and that thermodynamic changes dominate changes in hydrologic cycle extremes (see some select references below). That is not to say that the minor variation presented in this manuscript (the focus on extreme ARs in particular) are irrelevant; rather, it seems more appropriate to convey these incremental advances in a discipline-specific journal.

The second of the three takeaways ("extreme AR frequency doubles in future climate simulations") has the potential to be high impact, as I am not aware of any similar result having been presented in the literature. If the paper were revised in such a way that this statement is clearly robust with respect to the various important sources of uncertainty, the manuscript as a whole might be of higher impact. However, addressing this would likely involve substantially rewriting the paper, as it would likely require analysis of HighResMIP simulations; I can't see how the authors would be able to do this without essentially writing a new paper altogether.

References related to high resolution and extremes:

* Wehner, M., K. A. Reed, D. Stone, W. D. Collins, and J. Bacmeister, 2015: Resolution dependence of future tropical cyclone projections of CAM5.1 in the US CLIVAR Hurricane Working Group idealized configurations. *J. Clim.*, 150212130600007, <https://doi.org/10.1175/JCLI-D-14-00311.1>.

* Rauscher, S. A., T. A. O'Brien, C. Piani, E. Coppola, F. Giorgi, W. D. Collins, and P. M. Lawston, 2016: A multimodel intercomparison of resolution effects on precipitation: simulations and theory. *Clim. Dyn.*, 47, 2205–2218, <https://doi.org/10.1007/s00382-015-2959-5>.

* Scher, S., R. J. Haarsma, H. de Vries, S. S. Drijfhout, and A. J. van Delden, 2017: Resolution dependence of extreme precipitation and deep convection over the Gulf Stream. *J. Adv. Model. Earth Syst.*, 9, 1186–1194, <https://doi.org/10.1002/2016MS000903>.

References related to thermodynamic scaling and hydrological extremes:

* Emori, S., 2005: Dynamic and thermodynamic changes in mean and extreme precipitation under changed climate. *Geophys. Res. Lett.*, 32, L17706, <https://doi.org/10.1029/2005GL023272>.

* Gao, Y., J. Lu, L. R. Leung, Q. Yang, S. Hagos, and Y. Qian, 2015: Dynamical and thermodynamical modulations on future changes of landfalling atmospheric rivers over western North America. *Geophys. Res. Lett.*, 42, 7179–7186, <https://doi.org/10.1002/2015GL065435>.

* Prein, A. F., and L. O. Mearns, 2021: U.S. Extreme Precipitation Weather Type Increased in Frequency During the 20th Century. *J. Geophys. Res. Atmos.*, 126, 1–18, <https://doi.org/10.1029/2020JD034287>.

We admit the 1st and 3rd key points in the previous manuscript may be weak in novelty. In the revised manuscript, new datasets and new analyses were added to strengthen the original and novel findings that may be of interest to broad communities.

Firstly, EAR analyses were extended to a broad variety of climate models including LR CMIP6 and HighResMIP. Compared to ERA5, EAR IVT in LR CESM are underestimated by ~50% (Revised **Fig. 1** and **Fig. R2** above) and the underestimates of EARs in HighResMIP remain as high as ~40% (**Fig. S2** and **Fig. R2** above), due to the restricted eddy-resolving capability in either the atmosphere or the ocean

component, or both (see model details in Methods and detailed descriptions of Fig.1 in the revision). A ubiquitous negative EAR bias is confirmed in current climate simulations. In contrast, EAR simulations in “eddy-resolving” HR CESM are substantially improved, providing a unique opportunity to examine the response of EARs to anthropogenic warming that may be severely biased in previous climate simulations.

We agree the resolution dependence of simulated extremes has been discussed in numerous existing studies (Wehner et al. 2015; Rauscher et al. 2016; Scher et al. 2017) and it has been well recognized that increased model resolution will benefit the simulation of extremes, which might be one of the motivations to develop HighResMIP. However, as can be seen, EAR simulations are still severely biased even in HighResMIP. Long-term “eddy-resolving” climate simulations for both the ocean and the atmosphere were not available before HR CESM was reported in the manuscript, which was demonstrated to significantly reduce the bias and therefore able to provide more reliable EAR projections.

With new support from LR CMIP6 and HighResMIP, the 1st key point we want to address in the revised manuscript is: “*EARs are severely underestimated in current ‘non-eddy-resolving’ climate models, casting significant uncertainties on their projections. This problem is largely solved in the ‘eddy-resolving’ HR CESM.*” To the best of our knowledge, this is the first time that such a severe bias of EAR simulation in current climate models is reported and quantified, and a substantial improvement in

“eddy-resolving” coupled climate simulations is shown at a global scale, which possibly leads to significant improvement on EAR projections.

Secondly, we agree that the dominant role of the thermodynamic changes in determining AR response under global warming has been widely reported at regional scales and we prove that similar mechanisms can be applied at a global scale in our study. For the novelty concern, this point was removed from the major findings. The related figure was moved to the Supplementary (**Fig. S5**) and the corresponding section was cut down and revised to include detailed discussions on previous studies and to clarify the updates of our study (See **Lines 138-142**): “The predominant role of thermodynamic response in determining AR projection under warming has been widely discussed in many previous studies at regional scales^{11,19,29} and the possible negative role of dynamic response in affecting AR changes has also been noted in the North Pacific and Mediterranean^{28,30}. It is further proved in HR CESM that similar mechanisms can be applied at a global scale.”

In the revised manuscript, a new section about the changing relationship between ARs and storms was added. The strong coupling between AR and storms is crucial for AR dynamics. Under global warming, the pairing relationship between EAR and storms drops slightly (2%-5%, **Tab.S1**) and the storm intensity pairing with EARs is reduced (10%-20%, **Fig. 5c,g**). Moreover, future EARs tend to locate further away from the storm center (**Fig. 5h**). Generally, we found a reduced coupling between EARs and storms in a warming climate. This may have great implications for EAR prediction as the high internal variability induced by synoptic storms is the primary source that

undermines the predictability of EARs. The 3rd key point we want to address in the revised manuscript is: *“The coupling relationship between EARs and storms will be reduced in a warming climate, potentially influencing the predictability of future EARs.”* See more detailed discussion in the new section in the revised manuscript (**Lines 170-198**).

Last but not least, as shown in replies to your “major issues” above, a comprehensive evaluation with regard to model uncertainties, ARDT uncertainties and other possible sources of uncertainties (data resolution, IVT thresholds to define EARs, different warming levels and significance test) was conducted to verify the result robustness.

Specifically, extended EAR analyses in LR CMIP6, HighResMIP confirmed the ubiquitous underestimates (~40%-50%) of EARs in “non-eddy-resolving” climate models whilst the EAR simulations are significantly improved in the “eddy-resolving” HR CESM, compared to ERA5. It is also verified that the improvement of EARs in HR CESM is independent of data resolution as EARs of similar amplitude are observed when regridding the HR IVT onto the LR grid.

Two additional ARDTs from ARTMIP--IVT85% and IPART are applied in HR CESM. Although the absolute amplitude of detected EARs differs among different ARDTs, the projected EAR changes under global warming are highly consistent, all demonstrating a global doubling of EARs with similar spatial distribution, indicating the robustness of projected EAR changes in HR CESM.

Sensitivities of projected EAR changes to different warming levels and IVT thresholds are also testified in HR CESM. The trend analysis shows that EARs increase almost linearly with temperature warming and the estimated increasing rate of EAR IVT is ~25% per decade during 2030-2100. More intense amplifications under global warming may be found if EARs are defined using a higher IVT threshold and vice versa. Discussions on these uncertainties were included in the revision. See details in replies to your “major issues” above.

Togetherly, these new analyses verified the severe underestimates of EARs in current “non-eddy-resolving” climate models and the substantial improvement in HR CESM. It is also well evidenced that the projected EAR changes in HR CESM (“*a global doubling of EARs (>1250 kg/m/s) by the end of the 21st century*”) is robust.

In summary, new datasets and new analyses were included, and the manuscript was reorganized and a large body was rewritten. The *three key findings* in the revised manuscript are: 1) EARs are severely underestimated in current “non-eddy-resolving” climate models, casting significant uncertainties on their projections. This problem is largely solved in the “eddy-resolving” HR CESM. 2) There will be a global doubling of EARs, and a more concentrated tripling for the landfalling EARs by the end of the 21st century under RCP8.5 warming scenario. 3) The coupling relationship between EARs and storms will be reduced in a warming climate, potentially influencing the predictability of future EARs. We believe these revised findings are novel, robust and of great interest to the broad communities in climate science, and are worthwhile to be reported in *Nature Communications*.

Minor and/or Specific Feedback

* *Why RCP8.5 - not other scenarios? what are the implications of this?*

The RCP8.5 warming scenario refers to a future climate with a high-level greenhouse gas concentration and is commonly used in climate projections. At present, future simulation outputs in the “eddy-resolving” HR CESM are only available under RCP8.5 warming scenario. This was clarified in Methods (Lines 233-235): “Historical forcing is applied from 1850 to 2005 while RCP8.5 warming forcing (a high-level greenhouse gas concentration) is applied from 2006 to 2100³⁶⁻³⁸.”

** which ARDT was used -- it wasn't clear whether it was group's own or an existing one. The reference to Gimeno et al. (line 268) seems like it's intended to reference the ARDT used, but Gimeno et al. is just a short reference. Later in the paragraph (line 274), Xu et al. is referenced, but it isn't clear whether that reference is meant to justify the aspect ratio and isoperimetric quotient ratio or whether it's meant to indicate the ARDT that was used.*

We apologize for the unclear descriptions. The primary ARDT used in this study is based on the classic definition of ARs, which searches for long narrow features of IVT anomalies exceeding 250 kg/m/s using 6-hourly IVT ((as reviewed in Gimeno et al. 2014, hereafter IVT250). Additional geometric restrictions applied include a minimum length requirement of 800 km, a minimum length/width ratio of 2 and a minimum isoperimetric quotient ratio of 0.7 following a previous study (Xu et al. 2020). A minimum 24 hours duration restriction is also applied to the tracking to exclude short-lived moisture filaments. Detailed descriptions of this IVT250 method were added in

Methods (Lines 265-274). Also, additional two ARDTs from ARTMIP were applied and compared with the IVT250 to test result sensitivities to ARDTs in the revision (See Fig. S4 and detailed replies to your comments on “ARDT uncertainty” above).

** line 34: ARs are not necessarily always associated with ETCs (e.g., see Zhang and Ralph 2019, Zhang et al. 2021). Also note that this is a misstatement of Ralph et al.; in coming up with the AMS glossary definition, they intentionally declined to state whether or not ARs are always associated with ETCs (hence the language 'often associated with ETCs')*

Revised. Thanks. See Line 29.

** "characteristic PDFs" (e.g., line 167) <-- what is this? is it just a histogram? Or is it "PDF of characteristics"? I haven't heard this term before, and I am nominally in the same field of study as the authors of this manuscript.*

The term was replaced by “PDFs of AR characteristics” in the main text (Line 163). Fig.4 was revised and the term was not applicable in the new legend.

References

1. Reid, K. J., O'Brien, T. A., King, A. D. & Lane, T. P. Extreme Water Vapor Transport During the March 2021 Sydney Floods in the Context of Climate Projections. *Geophysical Research Letters* 48, (2021).
2. Xu, G., Ma, X., Chang, P. & Wang, L. Image Processing Based Atmospheric River Tracking Method Version 1 (IPART-1). *Geosci. Model Dev.* (2020).
3. Guan, B. & Waliser, D. E. Detection of atmospheric rivers: Evaluation and application of an algorithm for global studies: Detection of Atmospheric Rivers. *J. Geophys. Res. Atmos.* 120, 12514–12535 (2015).
4. Xu, G., Ma, X., Chang, P. & Wang, L. A Comparison of Northern Hemisphere Atmospheric Rivers Detected by a New Image-Processing Based Method and Magnitude-Thresholding Based Methods. *Atmosphere* 11, 628 (2020).
5. O'Brien, T. A. *et al.* Detection Uncertainty Matters for Understanding Atmospheric Rivers. *Bulletin of the American Meteorological Society* 101, E790–E796 (2020).

6. Ralph, F. M. *et al.* A Scale to Characterize the Strength and Impacts of Atmospheric Rivers. *Bulletin of the American Meteorological Society* **100**, 269–289 (2019).
7. Wehner, M. F. *et al.* The effect of horizontal resolution on simulation quality in the community atmospheric model, CAM5.1. *J. Adv. Model. Earth Syst.* **6**, 980–997 (2014).
8. Rauscher, S. A. *et al.* A multimodel intercomparison of resolution effects on precipitation: simulations and theory. *Clim Dyn* **47**, 2205–2218 (2016).
9. Scher, S., Haarsma, R. J., de Vries, H., Drijfhout, S. S. & van Delden, A. J. Resolution dependence of extreme precipitation and deep convection over the Gulf Stream. *J. Adv. Model. Earth Syst.* **9**, 1186–1194 (2017).
10. Lavers, D. A. *et al.* Future changes in atmospheric rivers and their implications for winter flooding in Britain. *Environ. Res. Lett.* **8**, 034010 (2013).
11. Payne, A. E. *et al.* Responses and impacts of atmospheric rivers to climate change. *Nat. Rev. Earth Environ.* **1**, 143–157 (2020).
12. Huang, X., Hall, A. D. & Berg, N. Anthropogenic Warming Impacts on Today's Sierra Nevada Snowpack and Flood Risk. *Geophys. Res. Lett.* **45**, 6215–6222 (2018).
13. Gao, Y. *et al.* Dynamical and thermodynamical modulations on future changes of landfalling atmospheric rivers over western North America. *Geophys. Res. Lett.* **42**, 7179–7186 (2015).
14. Polade, S. D., Gershunov, A., Cayan, D. R., Dettinger, M. D. & Pierce, D. W. Precipitation in a warming world: Assessing projected hydro-climate changes in California and other Mediterranean climate regions. *Sci Rep* **7**, 10783 (2017).
15. Lamarque, J.-F. *et al.* Global and regional evolution of short-lived radiatively-active gases and aerosols in the Representative Concentration Pathways. *Climatic Change* **109**, 191–212 (2011).
16. Lamarque, J.-F. *et al.* Historical (1850–2000) gridded anthropogenic and biomass burning emissions of reactive gases and aerosols: methodology and application. *Atmos. Chem. Phys.* **10**, 7017–7039 (2010).
17. Meinshausen, M. *et al.* The RCP greenhouse gas concentrations and their extensions from 1765 to 2300. *Climatic Change* **29** (2011).
18. Gmeno, L., Nieto, R., Vázquez, M. & Lavers, D. A. Atmospheric rivers: a mini-review. *Front. Earth Sci.* **2**, (2014).

REVIEWER COMMENTS

Reviewer #2 (Remarks to the Author):

In revising their manuscript, now titled "Extreme Atmospheric Rivers in a Warming Climate," Wang et al. have undertaken an impressively rigorous analysis of extreme atmospheric rivers in high-resolution atmosphere-ocean CESM, in a low-resolution counterpart, and in a number of HighResMIP simulations. The analysis in the revised manuscript is comprehensive in its treatment of uncertainty.

The authors have revised the manuscript appropriately in response to both my comments and those from another reviewer, resulting in a high-impact manuscript that I expect will ultimately be appropriate for publication in Nature Communications.

I only have a few remaining minor concerns and therefore recommend that the manuscript be accepted for publication pending minor revisions.

Note that my students again assisted with this review.

Minor concerns

Discussion of EAR changes in low-res simulations

The most notable of my major concerns relates to a point that would benefit from further discussion in the manuscript. One of the authors' key points relates to the ubiquitous low bias in EAR in non-eddy-resolving AOGCM simulations, which is quite evident from Figure 1; on the basis of this point, the manuscript seems to imply that eddy-resolving AOGCM simulations are necessary to study EARs.

While this may well be true, it is noteworthy that Figures 3b,c show that changes in EAR frequency and EAR IVT in the low-resolution simulations are comparable in relative magnitude to the changes evident in the HR-CESM simulations. This raises a question: could the 2050-2100 HR-CESM results be reproduced with low-resolution by simply bias-correcting the low-resolution results?

Another way of making this same point is that a bias in the state of a model's historical simulations doesn't necessarily imply that the model isn't useful for providing information about future climate change (sorry for the double negative there--couldn't figure out how to write that sentence without it). Points similar to this theme are ubiquitous in the literature, especially with respect to mean precipitation in climate models: see, for example, IPCC AR6 Chapter 8 (Douville et al, 2021; especially section 8.5.1).

The manuscript would benefit from further discussion on this point.

Douville, H., K. Raghavan, J. Renwick, R.P. Allan, P.A. Arias, M. Barlow, R. Cerezo-Mota, A. Cherchi, T.Y. Gan, J. Gergis, D. Jiang, A. Khan, W. Pokam Mba, D. Rosenfeld, J. Tierney, and O. Zolina, 2021: Water Cycle Changes. In Climate Change 2021: The Physical Science Basis. Contribution of Working Group I to the Sixth Assessment Report of the Intergovernmental Panel on Climate Change [Masson-Delmotte, V., P. Zhai, A. Pirani, S.L. Connors, C. Péan, S. Berger, N. Caud, Y. Chen, L. Goldfarb, M.I. Gomis, M. Huang, K. Leitzell, E. Lonnoy, J.B.R. Matthews, T.K. Maycock, T. Waterfield, O. Yelekçi, R. Yu, and B. Zhou (eds.)]. Cambridge University Press, Cambridge, United Kingdom and New York, NY, USA, pp. 1055–1210, doi: 10.1017/9781009157896.010.

Interpretation of 'global average EAR IVT'

Figure 3 and parts of the text (e.g., around line 114) discuss global average EAR IVT. My students and I were collectively confused about what this quantity is. My understanding is that EAR IVT is the IVT within extreme atmospheric rivers, which are defined in most places in the manuscript as ARs with max IVT greater than 1,250 kg/m/s and contiguous regions of IVT with magnitudes of at

least 250 kg/m/s (with some additional constraints). So what is the global average of this? Is it the average over all ARs, not considering non-AR areas? If so, how is it that values in Figure 3 go below 250 kg/m/s.

We're obviously missing something here, and I am guessing that this might be fixed by a more detailed explanation of what global average EAR IVT is.

Saturated colorbar

There are substantial regions in Figure 4 (especially Figures 4g and 4h, but also 4d and 4j) where the shaded regions are at the upper-end of the colorbar. Saturating the color bar like this should generally be avoided, or if it is unavoidable, the maximum value in the plot should at least be reported. My recommendation would be to expand the colorbar range to avoid having areas that saturate the colorbar range. I'll also note that it may not be necessary to have a blue-white-red colorbar for the climate change figures, since there are no areas in which a decrease is reported.

Figure 5 needs some TLC

- * The title in Figure 5e seems to be incorrect; I think that is supposed to be SH?
- * I've spent about 10 minutes on Figures 5c and 5g (with my students too) and we are still uncertain what the shading represents and what the contours represent. I think what would help is having the colorbars labelled.

Reply to referee #2

First, we would like to thank the referee again for the invaluable comments. We have carefully considered each of your comments (*listed as bold and Italic below*) and revised the manuscript accordingly. Below are point-to-point replies to your comments:

Summary:

In revising their manuscript, now titled "Extreme Atmospheric Rivers in a Warming Climate," Wang et al. have undertaken an impressively rigorous analysis of extreme atmospheric rivers in high-resolution atmosphere-ocean CESM, in a low-resolution counterpart, and in a number of HighResMIP simulations. The analysis in the revised manuscript is comprehensive in its treatment of uncertainty.

The authors have revised the manuscript appropriately in response to both my comments and those from another reviewer, resulting in a high-impact manuscript that I expect will ultimately be appropriate for publication in Nature Communications.

I only have a few remaining minor concerns and therefore recommend that the manuscript be accepted for publication pending minor revisions.

Note that my students again assisted with this review.

Minor concerns:

Discussion of EAR changes in low-res simulations

The most notable of my major concerns relates to a point that would benefit from further discussion in the manuscript. One of the authors' key points relates to the ubiquitous low bias in EAR in non-eddy-resolving AOGCM simulations, which

is quite evident from Figure 1; on the basis of this point, the manuscript seems to imply that eddy-resolving AOGCM simulations are necessary to study EARs.

While this may well be true, it is noteworthy that Figures 3b,c show that changes in EAR frequency and EAR IVT in the low-resolution simulations are comparable in relative magnitude to the changes evident in the HR-CESM simulations. This raises a question: could the 2050-2100 HR-CESM results be reproduced with low-resolution by simply bias-correcting the low-resolution results?

Another way of making this same point is that a bias in the state of a model's historical simulations doesn't necessarily imply that the model isn't useful for providing information about future climate change (sorry for the double negative there--couldn't figure out how to write that sentence without it). Points similar to this theme are ubiquitous in the literature, especially with respect to mean precipitation in climate models: see, for example, IPCC AR6 Chapter 8 (Douville et al, 2021; especially section 8.5.1).

The manuscript would benefit from further discussion on this point.

*Douville, H., K. Raghavan, J. Renwick, R.P. Allan, P.A. Arias, M. Barlow, R. Cerezo-Mota, A. Cherchi, T.Y. Gan, J. Gergis, D. Jiang, A. Khan, W. Pokam Mba, D. Rosenfeld, J. Tierney, and O. Zolina, 2021: Water Cycle Changes. In *Climate Change 2021: The Physical Science Basis. Contribution of Working Group I to the Sixth Assessment Report of the Intergovernmental Panel on Climate Change* [Masson-Delmotte, V., P. Zhai, A. Pirani, S.L. Connors, C. Péan, S. Berger, N. Caud, Y. Chen, L. Goldfarb, M.I. Gomis, M. Huang, K. Leitzell, E. Lonnoy, J.B.R. Matthews, T.K. Maycock, T. Waterfield, O. Yelekçi, R. Yu, and B. Zhou (eds.)]. Cambridge University Press, Cambridge, United Kingdom and New York, NY, USA, pp. 1055–1210, doi: 10.1017/9781009157896.010.*

Thank you for raising this point and we agree. As you pointed out, although the absolute value of EARs is severely underestimated in “non-eddy-resolving” LR climate

models for both historical (HIS) and future (RCP) simulations, the magnitude of the relative EAR changes between RCP and HIS is captured. This means that LR climate models may still provide valuable insights into future EAR projections if the simulation bias in their historical runs is corrected. A related discussion was added in the revision (Lines 130-135): “However, it is worthwhile to note that the magnitude of the relative EAR change projected by LR CMIP6 is comparable to that in HR CESM. This implies that LR climate models may still provide valuable information about future EAR projections if the systematic model bias in their historical simulations is precisely known and corrected, as also noted in previous mean precipitation projections²⁸.”

Interpretation of 'global average EAR IVT'

Figure 3 and parts of the text (e.g., around line 114) discuss global average EAR IVT. My students and I were collectively confused about what this quantity is. My understanding is that EAR IVT is the IVT within extreme atmospheric rivers, which are defined in most places in the manuscript as ARs with max IVT greater than 1,250 kg/m/s and contiguous regions of IVT with magnitudes of at least 250 kg/m/s (with some additional constraints). So what is the global average of this? Is it the average over all ARs, not considering non-AR areas? If so, how is it that values in Figure 3 go below 250 kg/m/s.

We're obviously missing something here, and I am guessing that this might be fixed by a more detailed explanation of what global average EAR IVT is.

We apologize for the lack of clarity in previous descriptions. The global averaged EAR IVT is computed by averaging the accumulated EAR IVT across all the red boxes in Fig.2d. As stated in the manuscript, the accumulated EAR IVT is computed to take account of the combined impact of the frequency and intensity of EARs (see Methods for details, **Line 294-302**). Specifically, at each grid point, the EAR IVT shown in Fig. 2 is computed as $\frac{\sum_{AR=1}^n IVT}{N}$ (where IVT is the intensity of an individual EAR, n is the number of EARs passing the computing grid, N is the maximum EAR occurrence number among all the computing grids used for normalization). Although the individual EAR IVT is all above 250 kg/m/s, it should be noted that EARs are distributed over space and do not perfectly align. Therefore, it is possible that only a fraction of EARs overlapped at any given computing grid point ($n < N$), leading to reduced IVT values (possibly below 250 kg/m/s in certain areas) compared with individual EAR IVT.

In the revision, “global averaged IVT” was replaced by “global averaged accumulated EAR IVT” in the legend of Fig. 3 (**Lines 512,513**) for clarity, and the calculation details were also added in the figure legend (**Lines 517-518**): “The global averaged EAR IVT/frequency is computed by averaging the corresponding values in red boxes outlined in Fig. 2d”. Moreover, the reason for the reduced EAR IVT values was explained in detail in Methods (**Lines 302-306**): “Note that although the individual EAR IVT is all above 250 kg/m/s, EARs are distributed over space and do not perfectly align. Therefore, it is possible that only a fraction of EARs overlapped at any given computing grid point ($n < N$), leading to reduced IVT values (Fig. 2d, 3c) compared with individual EAR IVT.”

Saturated colorbar

There are substantial regions in Figure 4 (especially Figures 4g and 4h, but also 4d and 4j) where the shaded regions are at the upper-end of the colorbar. Saturating the color bar like this should generally be avoided, or if it is unavoidable, the maximum value in the plot should at least be reported. My recommendation would be to expand the colorbar range to avoid having areas that saturate the colorbar range. I'll also note that it may not be necessary to have a blue-white-red colorbar for the climate change figures, since there are no areas in which a decrease is reported.

Following your suggestion, the corresponding colorbar range was expanded and the maximum values were also labeled in the figure for clarity (see revised **Fig. 4**). A test plot using white-red colorbar is shown below (**Fig. R1**), which produces similar results as in **Fig. 4**. Throughout the manuscript, all the difference maps are plotted using the blue-white-red colorbar to represent the full range response (e.g. there are both positive and negative values in Fig. 5 and Fig. S5). To be consistent with other figures and to avoid any confusion that negative values may be intentionally dropped for general readers, we think it is helpful to keep the blue-white-red colorbar in Fig. 4.

Fig. R1 | Same as Fig. 4, but using white-red colorbar in the difference maps.

Figure 5 needs some TLC

*** The title in Figure 5e seems to be incorrect; I think that is supposed to be SH?**

*** I've spent about 10 minutes on Figures 5c and 5g (with my students too) and we are still uncertain what the shading represents and what the contours represent. I think what would help is having the colorbars labelled.**

We apologize for the typo and the unclear description. The title in **Fig. 5e** was revised. In **Fig. 5c,g**, the contours represent SLP gradient (∇SLP) in HR-HIS and the shading indicates ∇SLP difference between HR-RCP and HR-HIS. The figure titles and captions (**Line 536**) of **Fig. 5c,g** have been revised accordingly to clarify these information.